# Preoperative Cardiovascular Assessment of the Renal Transplant Recipient: A Narrative Review

**DOI:** 10.3390/jcm10112525

**Published:** 2021-06-07

**Authors:** Prasanti Alekhya Kotta, Madhivanan Elango, Vassilios Papalois

**Affiliations:** 1Department of Medicine, King’s College London, London SE5 9NU, UK; alekhya.kotta@kcl.ac.uk; 2Department of Surgery and Cancer, Imperial College London, London SW7 2AZ, UK; vassilios.papalois@nhs.net

**Keywords:** kidney, transplantation, cardiovascular optimisation, coronary artery disease

## Abstract

Patients with end-stage kidney disease (ESKD) have a high prevalence of cardiovascular disease; it is the leading cause of death in these patients and the optimisation of their cardiovascular health may improve their post-transplant outcomes. Patients awaiting renal transplant often spend significant amounts of time on the waiting list allowing for the assessment and optimisation of their cardiovascular system. Coronary artery disease (CAD) is commonly seen in these patients and we explore the possible functional and anatomical investigations that can help assess and manage CAD in renal transplant candidates. We also discuss other aspects of cardiovascular assessment and management including arrhythmias, impaired ventricular function, valvular disease, lifestyle and pulmonary arterial hypertension. We hope that this review can form a basis for centres hoping to implement an enhanced recovery after surgery (ERAS) protocol for renal transplantation.

## 1. Introduction

Cardiovascular disease (CVD) is the leading cause of morbidity and mortality among those with end-stage kidney disease (ESKD) [1] and kidney transplant recipients [2]. All major types of cardiovascular disease including coronary artery disease (CAD), valvular heart disease, arrhythmias, and pulmonary hypertension are prevalent among kidney transplant candidates.

The burden of cardiovascular disease will continue to increase as candidates on the kidney transplant waiting lists get older and more comorbid. In 2011, 62% of kidney transplantation candidates were over 50 years of age compared with 28.7% of kidney transplantation candidates in 1991 [3].

The imbalance between the availability of organs and the number of patients on the transplant waiting list necessitates objective selection criteria to guide transplant candidacy decisions. Preoperative cardiovascular assessment can be utilized to identify suitable candidates who are at a lower risk of adverse events and guide management to optimise transplantation outcomes.

Our review aims to highlight important points for a clinician to consider while performing a comprehensive, holistic cardiovascular screening assessment starting from the bedside assessment to the utilization of novel, highly sophisticated screening methods such as cardiac positron emission tomography (PET). We aim to provide a narrative review on the cardiovascular assessment and management of CAD, arrhythmias, heart failure, pulmonary hypertension, valvulopathies and lifestyle factors. This is however a very broad topic and we are not able to cover every issue relating to the cardiovascular system. Other complex issues including hypertension, right ventricular overload syndrome and hemodynamic monitoring are not covered in this review. We do include a summary of the most recent 2020 Kidney Disease: Improving Global Outcomes (KDIGO) Clinical Practice Guideline [4] and discuss the results of the groundbreaking ISCHEMIA-CKD trial [5]. We discuss the value of each screening assessment especially in the context of CKD, controversies in pre-transplant cardiac screening, inconsistencies between guidelines and the advantages and limitations of different screening strategies with the aim to help the reader make informed decisions about pre-operative cardiac screening and cardiovascular disease management.

## 2. Screening for CAD in Kidney Transplant Candidates

The case for CAD screening in kidney transplant candidates originates from the concept that chronic kidney disease (CKD) is a strong risk factor for developing coronary artery disease (CAD); as the glomerular filtration rate (GFR) declines below 60 mL/min/1.73 m^2^, the probability of developing CAD increases linearly [6]. CAD is a major cause of morbidity and mortality among kidney transplant candidates and the prevalence of CAD remains high after kidney transplantation [7]. Patients with angiographically significant CAD are at an increased risk of major adverse cardiac events (MACEs) and increased risk of mortality, within certain subgroups of patients such as those with more proximal CAD at a higher risk for MACE [3,7,8]. In addition, patients with CKD and ESKD also have worse outcomes following MACE compared to non-CKD counterparts [3,9].

In addition, many CKD patients remain asymptomatic despite developing severe CAD and there is a high prevalence of silent myocardial ischemia owing to factors such as diabetic and uremic neuropathy in CKD patients [9,10]. Benett et al. [11] examined 11 asymptomatic diabetic ESKD patients who voluntarily underwent coronary angiography and found multivessel CAD in all patients. Weinrauch et al. [12] examined 21 ESKDs with Type 1 diabetes with no clinical or ECG evidence of CAD and found that 50% had CAD and 38% had significant CAD. Dyspnea on exertion is also less specific for angina as it may be secondary to anaemia, volume overload, or metabolic acidosis in patients with CKD [9]. CAD amongst patients with CKD is also not universal: as many as 50% to 70% of patients with advanced CKD do not have obstructive CAD [13].

However, screening for occult CAD among kidney transplant candidates is not straightforward and there are significant disparities between the different guidelines (Table 1).

In addition, whether screening improves transplant outcomes or survival is uncertain and there is a risk that screening can lead to harm, unnecessarily subject candidates to invasive procedures and delay or exclude patients from transplantation [16,18,19]. There is also no evidence that pre-emptive coronary revascularisation improves outcomes in asymptomatic patients with stable CAD [5]. De Lima et al. [20] in their retrospective study of 1696 kidney transplant candidates found that screening led to a significant number of non-invasive and invasive tests. They also found that the identification of CAD was predictive of post-transplantation coronary events but not mortality and found that intervention did not alter survival. Dunn et al. [21] in their propensity score-matched cohort analysis of 17,304 kidney transplant recipients found that routine cardiac stress testing was not associated with a difference in the rates of death and myocardial infarction within 30 days of transplantation. As a result, the subject of pre-transplant cardiac screening is controversial with some calling for it to be abolished [22]. They argue that negative results can provide false reassurance and positive results can lead to further, sometimes invasive, investigations with associated costs, delay to treatment, risk of radiation [23] and unnecessary intervention [24].

However, in the setting of transplantation, screening can guide decisions about transplant candidacy [6] and direct pre-transplant optimisation. Screening can identify individuals with a high burden of CAD that confer poor prognosis and there is evidence, presented below, that screening tests can guide prognostication. Though, given the uncertain and controversial subject, the risk of publication bias towards studies that found significant results should be taken into account. Transplant candidates spend many months and years on the transplant waiting list and screening can be used to monitor for the development of CAD, initiate treatment and maintain medical fitness. Perioperative events can severely impact transplanted kidney function and with pre-operative optimisation, it is hoped (but not proven) that treatment invoked by screening may prevent perioperative events and improve long-term outcomes.

Methods of screening will now be discussed with the aim to help clinicians identify appropriate screening strategies in a patient and centre-specific manner. It is important to be conscious of the limitations of various screening methods, especially in the context of CKD.

## 3. Methods of Screening

Screening modalities can be categorized into history and physical examination, risk prediction, blood tests, electrocardiograms, functional status evaluation, non-invasive strategies and invasive strategies.

## 4. History and Physical Examination 

The patient history can be used to identify symptoms of CAD (anginal symptoms, exercise intolerance, shortness of breath) and screen for CAD risk factors (Table 2). It can be used to assess the patient’s general health, exercise tolerance, family history, comorbidities and quality of life. The physical examination can be used to assess for the presence of peripheral arterial disease, anaemia, hypertension or hypotension, abdominal obesity, heart failure, valvular heart disease, hypertrophic cardiomyopathy or arrhythmias. Limitations include the fact that typical CAD symptoms are less common in patients with CKD and patients may learn to avoid chest pain or SOB by not exerting themselves to their limit [25].

## 5. Risk Prediction Scores

Risk prediction scores such as the Framingham risk score commonly underestimate the risk of CAD in CKD patients [26]. Silver et al. [26] in their retrospective study of 956 kidney transplant recipients found that the Framingham risk score substantially underestimated MACE (actual-to-predicted event ratio 1.2–8.4 in different subgroups, all *p* < 0.0001) and found in their multivariate Cox modelling that only the Framingham risk score ≥ 10% and eGFR predicted MACE. The addition of other variables including C-reactive protein (CRP), uric acid and urine albumin-to-creatinine ratio was not found to increase the prediction of MACE. The greatest underestimation of risk occurred in patients with preexisting ischemic heart disease, diabetes and smoking history. Several other composite risk scores have been developed, but few have been externally validated [27].

## 6. Biomarkers 

Patients are known to have elevated baseline values of creatinine kinase (CK), creatinine kinase myocardial band (CK-MB) and cardiac troponin in advanced CKD in the absence of acute coronary syndrome (ACS) [6,9]. Regardless, elevated troponin T (TnT) and troponin I (TnI), both in the presence and absence of cardiac ischemia, are associated with increased all-cause and cardiovascular mortality in CKD and severe atherosclerotic CAD is more common among ESKD patients with elevated TnT [28]. 

In patients on dialysis, the sensitivity of high-sensitivity TnI for diagnosing MI remained high but specificity reduced [29]. There is minimal variability in high-sensitivity TnT in stable dialysis patients so a routine test to establish a baseline TnT value could improve the diagnosis of acute coronary syndrome [30].

## 7. Proteinuria

Studies have found proteinuria to be predictive for cardiovascular disease and associated with mortality and morbidity [31]. In one study, a higher urinary albumin concentration increased the risk of cardiovascular death after adjusting for other cardiovascular risk factors [32]. Bello et al. demonstrated that proteinuria at each stage of CKD was associated with a higher risk of cardiovascular disease [33]. These studies suggest a role for proteinuria in the pre-transplant setting to risk-stratify patients and identify those at an increased risk for cardiovascular disease.

## 8. Electrocardiography (ECG)

An abnormal ECG is predictive of cardiac death in kidney transplant candidates [25]. Changes on ECG such as pathological Q waves, ST-segment depression or elevation, T wave inversion, and bundle branch blocks were predictive of CAD with a sensitivity of 77% and specificity of 58% [34]. However, exercise ECG had a sensitivity of only 35% [34] with less than half of dialysis patients reaching target heart rate secondary to poor exercise tolerance.

Structural changes such as left ventricular hypertrophy (LVH) and arrhythmias can also be identified on ECG. Serial ECGs allow for the detection of new abnormalities and timely investigation and management. Ambulatory ECG rarely adds diagnostic or prognostic information that cannot be derived from stress testing. 

## 9. Functional Status Evaluation 

Pre-transplant poor physical function and low physical activity [35] are associated with worse outcomes during and after transplantation [36]. A cohort study of 540 patients found an association between low physical activity and increased risk of cardiovascular and all-cause mortality in kidney transplant recipients [37]. Rosas et al. in their prospective cohort study of 507 kidney transplant recipients, found that physical activity at the time of kidney transplantation is a strong predictor of all-cause mortality [35]. There is also growing evidence that exercise training can benefit kidney transplant recipients [38,39].

However, in clinical practice and studies on physical activity in kidney transplant candidates, there is not a standardised approach to functional status assessment [36]. There is also a lack of consensus on the management of poor functional reserve and at what point the risk of transplantation outweighs benefits.

The ideal functional status assessment tool evaluates several aspects of physical functioning, guides risk stratification and predicts outcomes. Assessment tools should be objective, easy to administer and reproducible. Today there are more than 75 functional status assessment tools, some of the most frequently used tools that have an evidence base in the transplant setting are discussed in Table 3. 

## 10. Exercise-Based Stress Assessment 

Exercise-based stress has several limitations when used in CKD patients. Patients with CKD often have poor exercise tolerance and frequently cannot achieve diagnostic exercise levels [6]. In patients with diabetes, a common co-morbidity in patients with CKD, autonomic dysfunction blunts heart rate and blood pressure response to exercise, limiting the utility of exercise-based stress [3].

## 11. Functional Non-Invasive Imaging

Functional non-invasive imaging is the most commonly used imaging modality to screen for CAD in kidney transplant candidates [49] and examples include echocardiography, myocardial perfusion scintigraphy (MPS) using single-photon emission computed tomography (SPECT) or positron emission tomography (PET), and cardiac magnetic resonance (CMR). It can include the detection of myocardial ischemia through exercise-induced stress; inotrope (dobutamine) induced wall-motion abnormalities detected during stress CMR or stress echocardiography; the vasodilator (adenosine or dipyridamole) induced perfusion abnormalities detected through SPECT, PET, contrast CMR or contrast echocardiography.

Common advantages of functional imaging are that it also permits the simultaneous assessment of chamber sizes, systolic and diastolic cardiac function and valvular disease. Studies show that functional imaging has good prognostic value in CKD patients; abnormal functional imaging results predict mortality and adverse outcomes [50]. A meta-analysis of 52 studies with 7401 participants found that non-invasive tests such as SPECT and stress echocardiography are as good as coronary angiography at predicting future adverse cardiovascular events in advanced CKD patients [50]. Functional imaging can be used to risk-stratify patients, identify high-risk patients that require further investigation and guide pre-transplant management of patients with CAD [51]. Trials have also shown that functional imaging tests have been associated with fewer referrals for downstream invasive coronary angiography (ICA) compared with a strategy relying on anatomical imaging [52].

Common limitations include the fact that structural changes in the heart for example LVH, LV dilation and fibrosis are common in CKD patients and can confound the detection of wall motion abnormalities and perfusion defects. In addition, some CKD patients are on antianginals such as beta-blockers and calcium channel blockers which can affect the response to pharmacological stressors [53,54,55], and arteriovenous fistulas (AVFs) used for haemodialysis can affect response to vasodilators [55,56]. Furthermore, in patients with diabetes, diffuse CAD in all major coronary arteries is frequently present, resulting in an absence of detectable perfusion abnormalities. Many of these issues can be overcome by using newer techniques such as PET imaging which permit the quantitative measurement of absolute myocardial blood flow (MBF) and flow reserve.

There will now be a discussion of important points to consider with each form of functional non-invasive imaging and these are summarized in Table 4.

## 12. Echocardiography 

Resting transthoracic echocardiography can identify regional wall motion abnormalities, decreased left ventricular (LV) function and increased LV size which may all indicate CAD [52]. It is widely available and sensitivity can be improved by the addition of a contrast [57]. In patients with hypertension and/or diabetes, findings such as reduced coronary sinus flow may predict CAD with good sensitivity and specificity [58].

Dobutamine stress echocardiography (DSE) works by dobutamine induced regional wall motion abnormalities and systolic dysfunction in the presence of underlying perfusion abnormalities. It is preferred in transplant candidates who have low blood pressure or reactive airway disease. Dobutamine stress echocardiography has good prognostic value and an abnormal dobutamine stress echocardiography scan is predictive of MACEs and cardiovascular and all-cause mortality [25,50,51]. Bergeron et al. [59] found that the percentage of ischemic segments on dobutamine stress echocardiography predict mortality. There is no radiation exposure, it is safe to use in CKD patients and is the cheapest functional imaging modality. However, limitations of echocardiography can include reduced accuracy due to poor acoustic windows in certain groups of patients due to obesity or tachycardia.

## 13. Single-Photon Emission Computed Tomography (SPECT)

Myocardial perfusion scintigraphy (MPS) can utilize SPECT, a nuclear imaging test that uses radioactive tracers to trace blood flow and cardiac perfusion. It can be utilised in patients with uncontrolled blood pressure or arrhythmias [50]. Dipyridamole is the typical pharmacological stressor used and works by increasing adenosine levels, causing vasodilatation. However, in patients with CKD, higher basal adenosine levels attenuate the detection of stressor induced perfusion abnormalities [25]. In addition, the common utilisation of anti-anginal and antihypertensive medicines by CKD patients, as described above, reduces sensitivity further. Another limitation is that attenuation correction is required to correct artefacts, resulting in low image quality. There is also considerable radiation exposure but it is safe to use in CKD patients. 

Prognostically, an abnormal SPECT scan nearly doubles the risk of death in CKD patients [6] and a normal SPECT is associated with a relatively low risk of future adverse events [50,51] It is widely available and the quantification of blood flow is now possible, overcoming limitations of subjective interpretation of flow abnormalities.

## 14. Cardiac Magnetic Resonance (CMR)

CMR can identify cardiomyopathy, cardiac remodeling, infarction, myocardial fibrosis and myocardial infiltration which have important prognostic value [13]. Cardiac MRI evaluates for myocardial ischemia by either vasodilator induced perfusion abnormalities (require gadolinium-based contrast agents (GBCAs)), or dobutamine induced wall-motion abnormalities (does not require GBCAs). Dobutamine stress MRI has higher technical feasibility and diagnostic accuracy compared to dobutamine stress echocardiography: Dundon et al. evaluated dobutamine stress MRI in a pre-kidney transplant population and reported sensitivity of 100% and specificity of 89% for detecting angiographically significant CAD [60]. However, limitations include the fact that GBCAs have been linked to nephrogenic systemic fibrosis and there is a class warning against their use in patients with advanced CKD [13]. Hence, the diagnostic and prognostic value of stress MRI perfusion studies has not been tested in patients with CKD. The newer macrocyclic GBCAs have been deemed substantially safer than linear-structured GBCAs and may allow contrast CMR in renal transplant candidates [61]. 

## 15. Positron Emission Tomography (PET)

Positron emission tomography is a nuclear imaging modality that uses radioactive tracers to assess perfusion and metabolism. PET permits quantitative measurements of rest and stress myocardial blood flow (MBF) as well as absolute and relative flow reserve, which have high diagnostic and prognostic values for CAD [13]. Quantitative measurements permit better recognition of focal epicardial and diffuse microvascular disease. However, PET imaging is expensive and currently only available in a few, specialist centres. 

## 16. Anatomical Imaging 

Anatomical imaging modalities involve direct visualization of the coronary arteries either non-invasively for example through CT or MRI or invasively through coronary angiography.

## 17. Coronary Artery Calcium Score (CACS)

A non-contrast coronary computed tomography angiography (CCTA) permits the identification and quantification of calcium in the coronary arteries and calculation of CACS, most often using the Agatson scoring system [62]. CACS is often used as a marker of coronary atherosclerosis burden and has several advantages in CKD patients compared to other anatomical scans: both a contrast coronary computed tomography angiography (CCTA) and invasive coronary angiography (ICA) have higher radiation exposure and require contrast infusion which has adverse effects in patients with CKD. 

The limitations of CACS include that in CKD patients, there is a higher prevalence of vascular calcification and a significant amount of calcium is deposited in the medial arterial layer, coexisting with subintimal atherosclerotic plaque calcification. This means that CACS may not be a true reflection of the overall atherosclerotic plaque burden in CKD patients. Multiple studies have shown a poor correlation between CACS and CAD on angiography in ESKD and the localization of calcium also does not have a strong correlation with the vulnerability of coronary plaques. Statin therapy also stabilizes plaques and increases the CACS. 

Despite this, a high CACS has been shown to predict obstructive CAD, MACEs, morbidity and mortality in patients with CKD although the cutoff score predictive of CAD is higher than in patients without CKD [7,55,63]. 

## 18. Coronary Computed Tomography Angiography (CCTA)

CCTA involves using an intravenous contrast to non-invasively evaluate the coronary arteries. CCTA has evolved to become a mainline investigation in the evaluation of CAD. New advancements have led to high diagnostic accuracy and high-resolution anatomical delineation of the coronary arterial wall and lumen. It has high sensitivity, with most studies reporting sensitivity greater than 90% but specificity is reduced in CKD due to the presence of medial wall calcification [13,57]. CCTA can provide information on the degree of coronary stenosis, total plaque volume, plaque characteristics, features of plaque vulnerability and when combined with newer techniques such as perfusion or fractional flow reserve, functional severity of stenosis can be also assessed [57]. CCTA characteristics of unstable plaques include low attenuation, spotty calcification, large plaque volume, and higher remodeling index compared to chronic stable plaques. CCTA is considered to be the preferred test in patients with a lower clinical likelihood of CAD: it is a good rule-out test, with a high negative predictive value and has higher accuracy when low clinical likelihood populations are subjected to examination.

The presence or absence of non-obstructive coronary atherosclerosis on CCTA provides prognostic information and can be used to guide preventive therapy. In support of the prognostic power of CCTA, Winther et al. [7], in their prospective study of 154 patients referred for cardiac evaluation before kidney transplantation, found that both CCTA and invasive coronary angiography had similar value in predicting MACE and only CCTA was predictive of all-cause mortality, highlighting that CCTA can act as an effective gatekeeper to invasive coronary angiography. 

## 19. Invasive Coronary Angiography (ICA)

ICA involves visualization of the coronary arteries through catheterization and injection of contrast directly into the coronary arteries. It is considered the gold standard for CAD detection but requires significant radiation and contrast exposure as well as procedural risks, including vascular and bleeding complications. It is reserved for patients with suspected CAD and inconclusive non-invasive testing; a high clinical likelihood of CAD and symptomatic CAD unresponsive to optimal medical therapy; angina at a low level of exercise or results of CAD screening investigations indicating a high event risk. Appropriate utilisation of non-invasive screening tests can identify patients who do not have CAD and can help avoid unnecessary invasive testing. Studies have found that the yield of an angiogram can only be as great as the clinical indication for the test: coronary angiography is high-yield for obstructive CAD when non-invasive testing is performed in a patient with strong clinical risk factors for CAD [64].

## 20. Evaluating the Principle of CAD Screening

In terms of evaluating the screening program, the WHO principles of screening advise that a screening program should have scientific evidence of effectiveness. Currently, there is no strong evidence that cardiac screening in kidney transplant candidates improves outcomes. The screening program should have mechanisms to minimize potential risks: it is currently not supported by discrepancies between clinical guidelines (Table 1). The overall benefits of screening should outweigh the harm, and this is yet to be proven in this setting [22].

## 21. Summary on Screening for CAD

A thorough history can be used to screen for symptoms of CAD, and CAD risk factors, identify family history of CAD and assess the patient’s general health. Biomarkers including elevated troponin levels and proteinuria have been shown to identify candidates at higher risk of cardiac events and other adverse outcomes. ECG can be used to identify features of underlying cardiovascular disease including CAD and structural heart disease. An abnormal ECG has been found to be predictive of cardiac death in kidney transplant candidates but the utility of exercise ECG is reduced in those with poor exercise tolerance. Pre-transplant poor functional status has been found to be associated with worse outcomes post-transplantation and functional status assessment tools can be used to identify individuals with poor physical function. Functional non-invasive imaging strategies include echocardiography, SPECT, CMR and PET imaging and some of these imaging techniques have been identified to have good prognostic value in CKD patients. Some limitations of functional imaging in CKD patients are that structural changes in the heart, arteriovenous fistulae and diffuse CAD, seen in some patients with CKD, limiting interpretation of these scans. Anatomical non-invasive imaging strategies include CACS and CCTA but the higher prevalence of medial layer vascular calcification confounds interpretation of these scans in CKD patients. Newer assessment strategies such as fractional flow reserve and plaque characterization improve diagnostic accuracy and studies have found CCTA to be predictive of MACE and all-cause mortality in kidney transplant candidates. Invasive coronary angiography is the gold standard for CAD detection but is associated with significant risks including radiation and contrast exposure as well as procedural risks. We believe that the above screening strategies can serve as an effective gatekeeper to invasive coronary angiography and can help avoid unnecessary invasive testing. This is our own general perspective on the investigation of coronary artery disease and this opinion should not replace established guidelines.

## 22. Management of Stable CAD in Kidney Transplant Candidates

This section will focus on the management of stable CAD in kidney transplant candidates. Management of CAD is complicated in CKD patients. The high comorbidity burden, concerns about side-effects from therapies and the under-representation of CKD patients in clinical trials lead to a sparse evidence base.

## 23. Medical Therapy for Stable CAD 

There is growing evidence on the efficacy of treating stable CAD with optimal medical therapy (Table 2). The ISCHEMIA-CKD (International Study of Comparative Health Effectiveness with Medical and Invasive Approaches) trial [5] is the first and only trial that investigated the treatment of stable CAD in patients with CKD. Before this trial, findings were extrapolated from trials conducted on non-CKD patients with very few CKD patients or based on observational data. The ISCHEMIA-CKD trial was a large randomized clinical trial (RCT) of 777 patients with CKD stages 4 to 5D. It compared an invasive strategy (coronary angiography and revascularisation with optimal medical therapy) to a conservative strategy (optimal medical therapy only). They found that in patients with stable coronary artery disease, moderate or severe ischemia determined by a positive cardiac stress test and eGFR < 30 mL/min/1.73 m^2^ (including dialysis), a more invasive strategy offered no additional benefit compared to optimal medical therapy (Table 5) in terms of death, nonfatal myocardial infarction, hospitalization for unstable angina, heart failure, resuscitated cardiac arrest and angina-related health status between the two groups [5,67]. However, the trial excluded patients with symptomatic coronary artery disease, heart failure, and recent acute coronary syndromes or who had an ejection fraction of less than 35%.

Kamran et al. [68] conducted a meta-analysis of six studies comparing revascularisation versus medical management of stable obstructive CAD in prerenal transplant patients. They found no difference in post-transplantation cardiovascular outcomes between the two strategies. The Coronary Artery Revascularisation Prophylaxis (CARP) trial [69] and the Dutch Echocardiographic Cardiac Risk Evaluation Applying Stress Echo-V (DECREASE-V) [70] conducted in the setting of elective vascular surgery also found that pre-emptive coronary revascularisation pre-operatively does not improve outcomes compared to optimal medical therapy. 

Guidelines recommend the continuing maintenance of cardioprotective medications including beta-blockers, statins, aspirin and angiotensin-converting enzyme inhibitors while waiting for kidney transplantation and in the perioperative period. Qiao et al. [71] found that discontinuing ACE-I or ARB therapy in patients with declining kidney function was associated with a higher risk of mortality and MACE but no statistically significant difference in the risk of ESKD. 

## 24. Revascularisation for Stable CAD

The indications for revascularisation in patients with stable CAD who receive optimal medical therapy are the persistence of symptoms despite medical treatment or to improve prognosis in those with high-risk CAD anatomy. Multiple randomized control trials [72,73,74,75,76] have shown that in patients with high-risk CAD anatomy (Table 6), revascularisation with percutaneous coronary intervention (PCI) or coronary artery bypass grafting (CABG) offers a survival benefit. A network meta-analysis of 100 trials with 93,553 patients comparing initial medical therapy with revascularisation (with either PCI or CABG) to initial medical therapy alone, reported improved survival among patients undergoing revascularisation compared to medical treatment alone [77]. A meta-analysis of seven RCTs demonstrated a survival benefit of CABG in patients with left main, triple-vessel and proximal LAD CAD compared to medical therapy [78]. The STICH trial demonstrated that CABG offers a survival benefit compared to medical therapy in patients with LV ejection fraction (LVEF) ≤ 35% [79].

In patients in whom revascularisation is recommended according to current clinical practice guidelines, this should occur before transplantation [16]. Coronary artery revascularisation using coronary artery stenting requires post-procedure dual antiplatelet therapy for at least six months. This increases the risk of bleeding, the risk of which is further elevated in patients with CKD. Guidelines also recommend delaying elective surgery for at least one year after insertion of a drug-eluting stent [4].

Bangalore et al. [80] conducted a propensity score-matched study of 5920 patients with CKD and compared outcomes with PCI and CABG. They found that PCI was associated with a lower short-term risk of death, stroke and AKI; a similar long-term risk of death but a higher risk of myocardial infarction (MI), and repeat revascularisation compared with CABG. However, in patients on dialysis, they found that PCI was associated with an increased long-term risk of death, higher MI and repeat revascularisation compared to CABG. ECS/EACTS guidelines [76] recommend using the Society of Thoracic Surgeons (STS) score and EuroSCORE II to assess the morbidity and mortality after CABG; the SYNTAX score to assess the anatomical complexity of CAD and long-term mortality and morbidity after PCI to guide the choice between CABG or PCI for revascularisation (Table 7). 

Management decisions should also take into account that all types of revascularisation are associated with a higher risk of morbidity and mortality in CKD compared with non-CKD patients and the long-term results are less favourable in CKD patients [80]. There are higher rates of short-term procedural risks, acute kidney injury (AKI), restenosis, stent thrombosis and bleeding among CKD patients. Dual antiplatelet therapy should be used for at least six months after the insertion of a drug-eluting stent with the associated increased risk of bleeding. Revascularisation may permanently exclude or delay patients from transplantation.

## 25. Summary of Management of Coronary Artery Disease

There are three main options for the management of coronary artery disease: medical therapy, percutaneous coronary intervention and coronary artery bypass grafting. We offer here our own general perspective on the management of coronary artery disease and this opinion should not replace established guidelines. There is no convincing evidence that pre-emptive revascularisation in those with stable coronary artery disease improves post-transplantation outcomes. The guidance therefore recommends optimal medical therapy in these patients: antiplatelets, statins, renin-angiotensin-aldosterone system downregulation and beta-blockade. In those with symptomatic coronary artery disease or particularly high-risk coronary vasculature, revascularisation with interventional cardiology or surgery is recommended. The specific decision between these two will depend on clinical, anatomical and technical aspects specific to each patient.

## 26. Arrhythmias and Sudden Cardiac Death in CKD and ESKD Patients 

Patients with CKD have an increased burden of cardiac arrhythmias and sudden cardiac death compared to those without CKD. Likely culprits include electrolyte abnormalities, structural changes in the heart, volume shifts, uremic milieu and ischemia common in CKD patients. Common types of arrhythmia in patients with CKD include reentrant, ventricular and ischemic arrhythmias, with atrial fibrillation (AF) being the most common type of arrhythmia in this patient population.

The presence of pre-transplantation cardiac arrhythmias is associated with an increased risk of morbidity, mortality and graft loss post-transplantation with up to 46% higher risk of mortality at five-year follow-up in patients with atrial fibrillation [81]. Although arrhythmias are associated with poor outcomes, there is limited testing available to predict the risk of sudden cardiac death and only a baseline ECG is currently recommended at the time of pre-transplant evaluation and there is insufficient evidence to support evaluation through ambulatory rhythm monitoring [82]. In a single-centre study of post-kidney transplantation ventricular arrhythmias: male gender, dialysis vintage and high preexisting CACS were associated with post-transplantation ventricular arrhythmias [83]. 

## 27. Management of Arrhythmias

Management of arrhythmias in CKD patients is challenging: arrhythmias are associated with a higher risk of mortality and ischemic stroke in CKD patients but these patients are also at a higher risk of bleeding. The decision to anti-coagulate, the choice of anticoagulant and the dose must be personalized to each patient after considering risks, benefits and alternatives with each patient. The decision between warfarin versus direct oral anticoagulants (DOACs) will need to consider the pharmacokinetics of drugs and renal dose adjustments [84]. KDIGO recommends a multidisciplinary approach involving the nephrologist, cardiologist, primary care physician and pharmacist when making these decisions [4].

## 28. Anticoagulation 

Evidence from RCTs support the safe use of warfarin and DOACs in CKD stages 1 to 3: DOACs, with superior safety profile and lower bleeding risk, are preferred in CKD stages 1 to 3 [85]. However, evidence is sparse and conflicting in more advanced stages of CKD (CrCl < 25–30 mL/min), with ESKD and dialysis in these patient groups being excluded from the large RCTs investigating the efficacy of warfarin and DOACs [85,86].

A subgroup analysis from the SPAF (Stroke Prevention in Atrial Fibrillation) III trials showed that the efficacy of warfarin was broadly similar in stage 3 CKD patients and patients without CKD [87]. However, caution is warranted on the use of warfarin in patients with more advanced CKD: a meta-analysis of 13 studies found that warfarin use in patients with ESKD had a neutral effect on the risk of ischemic stroke and all-cause mortality while it was associated with a significantly increased risk of major bleeding [88]. Other studies also report similar neutral effects of warfarin on the risk of ischemic stroke and thromboembolic events, with some even reporting increased risk of ischemic stroke [89,90,91]. A Swedish nationwide cohort study showed that patients with AF and CKD or ESKD would benefit from warfarin if stroke and bleeding risk factors are optimally managed and there is tight control on anticoagulation [92]. In CKD patients receiving warfarin, a time in the therapeutic range (TTR) of >70% independently predicted reduced risk of stroke, death and major bleeding but the risk of suboptimal TTR (<65%) was increased in the presence of CKD [93,94].

With the lack of evidence from RCTs on the efficacy of DOACs in advanced CKD, findings from pharmacological modelling have been used to guide practice. In Europe, reduced doses of rivaroxaban, apixaban and edoxaban have been approved to be used in patients with severe CKD (CrCl 15–29 mL/min) not on dialysis (Table 8) [95,96] with the European Society of Cardiology AF guidelines emphasizing that there are no randomized controlled trials on the use of DOACs in patients with severe CKD [96]. The US Food and Drug Administration also approved dabigatran in patients with CrCl 15–29 mL/min and the use of apixaban in patients with stable ESKD on dialysis [86]. The available data highlight that correct dosing of DOACs is essential: in a large cohort study, underdosing or overdosing of DOACs was associated with decreased safety [97]. The assessment of renal function before starting a DOAC and regular monitoring is advised.

Similar to findings in non-CKD patients, pharmacological rhythm control and rate control strategies are equivalent in their efficacy in terms of risks of heart failure, stroke and mortality. Catheter ablation is superior to pharmacological antiarrhythmic therapies in achieving freedom from AF recurrence [85]. The utility of implantable cardioverter defibrillators (ICDs) in patients with ESKD is not well known because these patients are often excluded in the large RCTs investigating their benefits. In a study of patients with ESKD, those who had an ICD inserted had an early lower risk of death compared to those who did not but overall mortality rates after implantation remained high and there was a high post-implantation infection rate [98]. Available data seem to suggest that the benefit of ICDs decreases with declining GFR [85].

## 29. Cardiac Function

Impaired ejection fraction (EF) is known to be a poor prognostic marker in heart failure with those with EF < 30% on HD having a nine times higher risk of mortality than those with EF > 60%. Multiple medical therapies including beta-adrenoreceptor antagonists (beta-blockers), angiotensin-converting enzyme inhibitors (ACEi), mineralocorticoid receptor antagonists and neprilysin inhibitors are known to reduce mortality in certain populations with heart failure. These medications should be optimised by cardiologists in patients with heart failure on the renal transplant waiting list. In those with symptoms of heart failure or a known diagnosis of heart failure, it is reasonable to recommend regular echocardiograms to monitor the ejection fraction. Arguably, the main role of these screening echocardiograms is to aid with the allocation of resources; given the high mortality of those with EF < 30%. 

## 30. Valvulopathies

Patients with ESKD are known to have a high incidence of valvular disease with valvular calcification being reported in more than 50% of patients in some studies [103,104]. We will mainly focus on aortic stenosis and mitral valve regurgitation in this review as these are the most common and most amenable to treatment. As part of the routine pre-operative screening, we recommend that patients undergo a TTE to look at myocardial and valvular function. This will allow the detection and assessment of valvular anomalies. It may be beneficial to have more regular screenings of those with mild/moderate valvular disorders as valvulopathies are known to have faster progression in patients on dialysis [105].

There is a multitude of evidence describing the criteria for medical or surgical intervention in these valvulopathies as well as good RCT evidence of their survival benefit over medical management. As patients are often on the waiting list for many months or years, the patient may benefit from these procedures as they primarily reduce the risk of cardiac events. There is also evidence to show that patients with these valve disorders have worse perioperative outcomes in non-cardiac surgery and a recent retrospective study showed that an aortic valve replacement had better perioperative outcomes than those with severe AS but no replacement [106,107,108,109]. Regarding the specific treatment of valve disorders, many guidelines exist for the treatment of valvular heart disease and reviewing these in detail would fall outside of the scope of this review [110]. There are also studies on the topic of aortic valve replacement in kidney transplant patients [111,112]. Both surgical and percutaneous options seem to have similar long term mortality but transcatheter aortic valve replacement (TAVR) may be associated with short initial LOS and cost of stay. 

## 31. Smoking

Cigarette smoking is known to worsen peri-operative outcomes with an increased incidence of primarily wound and pulmonary complications [113,114,115,116]. With renal transplantation in particular, smoking is known to increase malignancy, graft loss, cardiovascular disease and overall mortality [117,118,119,120]. Active tobacco smoking was associated with an eight-fold increase in graft loss in recipients aged over 60 (*p* < 0.001) in one study [121]. A cohort study from the US demonstrated an increased risk of graft failure at one, five and ten years post-transplantation in those who were smoking at the time of pre-transplant evaluation [122]. Most promising, those who had given up smoking at the time of evaluation have similar survival rates compared to non-smokers suggesting that smoking cessation may improve postoperative outcomes. Around one-quarter of patients on the transplant waiting list are current smokers and given the prolonged time spent on the waiting list, smoking cessation interventions should be targeted at these individuals with the possibility of much improved post-operative outcomes [117,123].

Possible interventions in this area range from behavioural to pharmacological. Two particularly intensive studies including weekly interventions and counselling sessions had a large effect on smoking cessation at the time of surgery (RR 10.76 95% CI 4.55–25.46) [114,124,125]. Studies offering less intensive therapy were found to be less effective with minimal if any change in clinical outcomes. Some trials have tested the effect of pharmacological therapy (nicotine replacement therapy or varenicline) with success. Wong et al. found that smoking cessation counselling along with varenicline increased abstinence from smoking up to 12 months following surgery [126]. A mixture of behavioural and pharmacological therapies should be explored with potential kidney transplant recipients to optimise their post-operative outcomes. 

## 32. Obesity

Obesity is associated with worse outcomes post-operatively with generally increased rates of wound complications and specifically graft loss and death in kidney transplantation [127,128]. As patients often spend a long time on the kidney transplant waiting list, this provides ample time for patients to undergo dietary counselling and a prehabilitation exercise program. Barberan-Garcia et al. conducted an RCT looking at a personalised prehabilitation pathway in patients undergoing major elective abdominal surgery. They showed that the prehabilitation group had enhanced exercise tolerance, reduced postoperative complications (RR 0.5, 95% CI 0.3–0.8) and significantly reduced ICU length of stay. Multiple interventions have been studied within the field of solid organ transplantation with most of the trials being conducted in cardiothoracic transplant candidates [129,130]. One study looked at implementing a prehabilitation program for patients on the renal transplant waiting list. There was a significant increase in physical activity post program and most interestingly, there was a decreased LOS in patients who went on to receive a transplant compared to matched controls ((five vs. ten days; RR = 0.69 95%CI 0.50–0.94) [39]. These data suggest that there is a role for a prehabilitation program to be established for patients on the kidney transplant waiting list with beneficial outcomes irrespective of whether participants go on to receive an organ.

There is also much interest in the possibility of bariatric surgery in either potential renal transplant candidates or post-renal transplant [131,132,133,134,135,136,137,138,139]. Several studies have shown that bariatric surgery is safe and feasible in patients with ESKD and kidney transplant recipients [133,138]. It is also known to be effective and cause a significant reduction in BMI [135]. This itself in renal transplant candidates may improve outcomes based on the association between high BMI and increased post-operative complications. There is also evidence of benefit from the kidney perspective with increased graft survival in kidney transplant patients undergoing bariatric surgery [132]. This area will need further research to delineate exactly which patients will benefit from surgery as well as whether the surgery should be before or after renal transplantation.

## 33. Pulmonary Hypertension

Pulmonary arterial hypertension (PAH) is becoming increasingly recognised in patients with ESKD with some studies reporting an incidence of 32% [140]. It is known to confer worse outcomes post-transplant but whether controlling PAH leads to better outcomes post-transplantation is unknown. There are often underlying disorders for which treatment is known to improve prognosis in the general population. We therefore think it is reasonable to screen for and control PAH in patients on the renal transplant waiting list while more studies are needed to definitively show that this is clinically beneficial to patients.

PAH is defined by a mean pulmonary artery pressure (PAP) of greater than 25 mmHg on right heart catheterisation (RHC). This is however an invasive procedure so TTE to measure estimated pulmonary artery systolic pressure (PASP) is commonly used as an initial technique to assess pulmonary pressures. Echocardiographic measurements have a sensitivity of 83% and specificity of 72% in detecting PAH so patients with an elevated PASP should then go on to have an RHC to confirm the presence of PAH [141].

It has variable aetiologies which are important to differentiate to allow for optimal management. The WHO classification has five categories:Intrinsic arteriopathy due results in increased pulmonary vascular resistance [142]: this can be seen in genetic disorders (hereditary PAH), idiopathically, connective tissue diseases (systemic lupus erythematosus and scleroderma), infections and portopulmonary syndrome. Some of these conditions can also result in renal impairment (systemic lupus erythematosus) while right heart failure due to PAH can lead to CKD due to venous congestion. This can be managed by treating any underlying condition as well as pulmonary artery vasodilators (endothelin receptor antagonists, prostacyclins, phosphodiesterase-5 inhibitors and ricioguat).Left ventricular dysfunction [143,144]: LV dysfunction will lead to pulmonary venous congestion resulting in a compensatory increase in right ventricular contraction to maintain blood flow. The increased pressures will lead to vascular remodeling in the long term which may result in persistent PAH even if left atrial pressures are reduced. This is the most common form of PAH in renal failure with a 30–50% incidence of LV dysfunction noted in some cohorts. Renal patients are at risk of both systolic dysfunction due to ischemia and cardiomyopathy as well as diastolic dysfunction due to myocardial stiffening from hypertension and diabetes mellitus.Hypoxic pulmonary vasoconstriction [145]: pulmonary vascular is unique in the human body in that there is hypoxic vasoconstriction allowing poorly oxygenated areas of the lung to shunt blood to well-oxygenated areas of lung reducing ventilation-perfusion mismatch. This can be problematic in primary lung disorders as hypoxemia will result in generalised pulmonary constriction and PAH. This is seen in many pulmonary disorders with reduced oxygenation including chronic obstructive pulmonary disease (COPD), obstructive sleep apnea (OSA) and fibrotic lung diseases. Obstructive sleep apnea has been identified in up to 60% of patients with ESKD partly due to the increased incidence of obesity. It is important to manage the underlying respiratory condition to prevent the progression of PAH.Chronic thromboembolic pulmonary hypertension (CTEPH) [146]: this is due to multiple pulmonary emboli forming in the pulmonary vasculature often due to multiple risk factors (obesity, inactivity, tobacco, vascular disorders). This can be treated with pulmonary endarterectomy although ESKD patients are likely to be high risk due to their co-morbidities or conservatively with anti-coagulation and riociguat.Multifactorial or unclear aetiology: these patients have PAH due to haematological disorders (haemolytic anaemia, myeloproliferative disorders), systemic disease (e.g Langerhans cell histiocytosis), metabolic disease or even unknown aetiology. In ESKD, a potential culprit could be a high-output arterio-venous fistula [146]. Retrospective studies have shown a high incidence of PAH in patients undergoing haemodialysis compared to peritoneal dialysis [147]. The increased cardiopulmonary flow (as seen in some patients with congenital heart disease) is hypothesised to lead to pulmonary vascular remodeling and PAH. In patients with potential high-output AVF, AVF occlusion testing with possible ligation could lead to amelioration of PAH.

The aetiology of PAH can be established by patient history (OSA), blood investigation (vasculitic screen, infection testing), pulmonary angiogram (CTEPH), TTE (LV dysfunction or valvulopathy) and RHC. RHC can be particularly useful in determining pulmonary capillary wedge pressure (PCWP), pulmonary vascular resistance (PVR) and cardiac output. With these parameters, it is possible to distinguish between group 1, group 2 and group 5 PAH and treat accordingly. Given the high prevalence of PAH and the fact that patients on the renal transplant waiting list are already screened with TTE, we believe that further screening with RHC in those with elevated PASP to diagnose and categorise PAH is warranted in patients on the kidney transplant waiting list. This can then be managed according to subtype with further studies needed to confirm whether treatment improves operative outcomes for patients.

## 34. Summary of Investigations

A summary of some considerations in the screening of renal transplant candidates is included in Table 9. We hope that our review highlights some important aspects for physicians to consider in the management of these complex patients.

## 35. Hints for Future Research 

Cardiovascular disease is a significant cause of morbidity and mortality in patients with ESKD but these patients are commonly excluded from major cardiovascular clinical trials. There remain many unanswered questions on optimal assessment and management of cardiovascular disease in patients with renal dysfunction. More randomized control trials incorporating patients with CKD and ESKD may benecessary.

There is an expansion of non-invasive assessment of the coronary artery architecture with a refinement of the criteria for intervention but more research is needed to guide decisions about suitable tests based on baseline cardiovascular risk and renal function. Some of these tools, including CMR and PET imaging, are yet to be thoroughly evaluated in patients with CKD and ESKD and the sensitivity and specificity of these tests in this patient population are yet to be identified. The optimal frequency of CAD screening while on the transplant waiting list is unknown, and this question is currently being investigated by the CARSK (Canadian-Australasian Randomised Trial for Screening Kidney) trial. Whether CAD screening leads to improved post-transplant outcomes is also unproven. Prospective randomized trials are needed to define which subsets of patients might benefit from pretransplant intensified medical management or from revascularisation.

Pre-transplant cardiac arrhythmias are associated with adverse outcomes but currently only a baseline ECG is recommended to evaluate for arrhythmias pre-operatively. Studies looking at the utility of ambulatory rhythm monitoring and other strategies to identify underlying arrhythmias would be beneficial. There is a lack of evidence from RCTs on the optimum dosing regimen, efficacy and safety of anticoagulants in advanced CKD. 

With structural heart disease, percutaneous options are becoming more widespread and further prospective studies are needed to study their benefit in the ESKD population and whether these improve post-transplant outcomes.

## 36. Conclusions

In conclusion, we hope that our review adds to the very complex issue of optimisation of the cardiovascular assessment and management of renal transplant candidates (summarized in Table 9), by reviewing some of the current evidence and highlighting some of the unmet needs. It is clear that while some therapies do lead to a survival benefit, more evidence is reviving the role of optimal medical management in these high-risk patients. We aim for our review to be a very initial starting point for a cardiovascular optimisation program for renal transplant patients as part of an ERAS pathway. We believe that this will require much clinician input as the cost, feasiblity, comprehensiveness and utility are just some of the factors that must be considered when creating a pathway. There is also potentially a need for future work to look at the survival benefit for some of these interventions.

## Figures and Tables

**Table 1 jcm-10-02525-t001:** Summary of main recommendations from international guidelines.

The 2020 Kidney Disease: Improving Global Outcomes (KDIGO) Clinical Practice Guideline [4]
Evaluate all candidates for the presence and severity of cardiac disease with history, physical examination, and ECG Candidates with signs or symptoms of cardiac disease should be referred to a cardiologist and undergo management before being considered for transplantation Candidates at high risk for coronary artery disease (CAD) or with poor functional capacity should undergo noninvasive CAD screening Asymptomatic candidates with known CAD should not be revascularised exclusively to reduce perioperative cardiac events Patients with asymptomatic, advanced triple vessel coronary disease; uncorrectable, symptomatic New York Heart Association (NYHA) Functional Class III/IV heart disease should be excluded from kidney transplantation unless they have an estimated survival which is acceptable according to national standards Asymptomatic candidates who have been on dialysis for at least two years or have risk factors for pulmonary hypertension should undergo echocardiography Patients with an estimated pulmonary systolic pressure greater than 45 mm Hg, severe valvular heart disease or myocardial infarction should be assessed by a cardiologist
American Society of Transplantation (AST) [14] (2002)
ECG and chest radiograph in all candidates Echocardiogram if the patient has left ventricular hypertrophy, congestive heart failure or myocardial dysfunction is suspected Noninvasive stress testing recommended for patients at “high risk” (diabetes, known ischemic heart disease, or 2+ risk factors: age ≥ 45 in men or ≥55 in women, ischemic heart disease in a first-degree relative, smoker, diabetes, hypertension, dyslipidemia, left ventricular hypertrophy) Coronary angiography for patients with a positive stress test Revascularisation before transplantation for patients with critical coronary lesions
American College of Cardiology/American Heart Association (ACC/AHA) [15] (2007)
Consider further cardiac evaluation in symptomatic patients Does not encourage further testing for patients who have no cardiac symptoms with a functional capacity of 4+ metabolic equivalents of tasks (METs) regardless of diabetes, history of CAD, or other traditional cardiac risk factors Consider noninvasive testing in asymptomatic patients with one or two clinical risk markers (ischemic heart disease, compensated or prior heart failure, diabetes, decreased kidney function, cerebrovascular disease) and poor functional capacity who require intermediate-risk noncardiac surgery if it will change management Recommendations for testing are stronger if 3+ clinical risk factors are present
American Heart Association/American College of Cardiology Foundation (AHA/ACCF) [16] (2012)
ECG and echocardiogram in all patients Noninvasive stress testing in kidney transplant candidates with no active cardiac conditions based on the presence of ≥CAD risk factors (diabetes, prior cardiovascular disease, >1 y on dialysis, left ventricular hypertrophy, age > 60 y, smoking, hypertension, and dyslipidemia) regardless of functional status Coronary angiography in patients who meet the criteria based on 2011 ACCF and AHA guidelines for coronary artery bypass graft surgery
European Renal Best Practice (ERBP) [17] (2015)
In asymptomatic low-risk candidates, basic clinical data, physical examination, ECG at rest, and chest x-ray are sufficient workup In asymptomatic high-risk patients, (older age, diabetes, personal or family history of cardiovascular disease), a standard exercise tolerance test and echocardiogram is recommended; in those with a true negative test result, further cardiac screening not indicated In candidates with high risk and a positive or inconclusive exercise tolerance test, further cardiac investigation for occult CAD with noninvasive stress imaging (myocardial perfusion or dobutamine stress echocardiography) is recommended Recommend coronary angiography in candidates with a positive test for cardiac ischemia

**Table 2 jcm-10-02525-t002:** Risk factors for CAD among kidney transplant candidates.

Known CAD Age Diabetes mellitus Dyslipidemia Hypertension Smoking history Family history Duration of dialysis treatment Left ventricular hypertrophy Left ventricular ejection fraction ≤ 40%

**Table 3 jcm-10-02525-t003:** Functional status assessment tools that can be used to evaluate kidney transplant candidates.

Assessment tool	Examples and Details	Advantages	Disadvantages
Self-reported physical assessment questionnaires	Short Form-36 Physical Function Scale, Instrumental Activities of Daily Living, Duke Activity Status Index (DASI), Physical Activity Scale for the Elderly (PASE)	Easy to conduct	Subjective, inaccurate reporting, difficult to use them longitudinally to quantify the improvement Studies have found the scores of the questionnaires to be associated with outcomes such as mortality in CKD patients [35,40,41]
Physical performance measures	Grip strength and 6-min walk test (6MWT)	Easy to use, low or no cost, time efficiency Objective In small ESKD cohort studies, better performance on the 6MWT was correlated with improved quality of life [42]	Assesses only specific functions and muscle groups Grip strength is significantly worse in the arm with arteriovenous fistula and older ESKD patients are already known to have lower grip strength. [43] The 6MWT can be unreliable due to variability resulting from changes in volume status and timing around dialysis (slow 6MWT if fluid overloaded) [44]
Short performance physical battery	Combines the use of three physical, lower-extremity performance measures: standing balance, walking speed, and chair stand tests	5–10 min to conduct Objective In a prospective study including 700 kidney transplant patients, the short performance physical battery score was associated with post-transplantation mortality [45]	Cannot be utilised in those with lower extremity abnormalities e.g., lower extremity amputations
Fried’s Frailty Phenotype Score	Five domains: weight loss, exhaustion, physical activity, grip strength, and walking speed	Measured frailty by Fried’s frailty phenotype scoring has shown a correlation with post-transplantation outcomes [46,47]	Unintentional weight loss and exhaustion—are subjective and self-reported American Society of Transplantation frailty assessment survey results show that the Fried’s frailty phenotype score was utilized by only 3.6% of the survey takers who reported assessing frailty for candidacy evaluation [48]

**Table 4 jcm-10-02525-t004:** Cardiac imaging to assess for coronary artery disease in CKD patients.

	Diagnostic Value	Prognostic Value	Benefits	Limitations	Radiation, Safety in CKD, Cost (US $)
**Echocardiography**	Variable numbers with most studies reporting moderate sensitivity in the mid-70s range and moderate specificity in the mid-80s range [13]	Abnormal dobutamine stress echocardiography is associated with an increased risk of MACEs, cardiovascular mortality and all-cause mortality [25,50,51] DSE is as good as ICA at predicting cardiovascular mortality and MACE [50]	Permits assessment of LV size and function, valve disease Widely available, bedside test Sensitivity can be improved by the addition of contrast [57]	Poor acoustic windows and tachycardia limit accuracy LV structural changes; antianginals and AVFs—common in CKD—can reduce sensitivity for wall motion abnormalities	No radiation exposure Safe in CKD $800 [57]
**SPECT**	Variable numbers with most studies reporting moderate sensitivity and specificity in the mid-70s range [13], some studies report sensitivities in the 90s [65]	Abnormal SPECT nearly doubles the risk of death in CKD patients [6]; a normal SPECT is associated with a relatively low risk of future adverse events [50,51] SPECT is as good as ICA at predicting cardiovascular mortality and MACE [50]	Permits assessment of LV function Widely available Quantification now possible, overcoming limitations of subjective interpretation	Attenuation correction is required to correct artefacts, resulting in low image quality LV structural changes; antianginal drugs and AVFs common in CKD, can reduce sensitivity for perfusion defects	Radiation exposure: 10–15 mSv Safe in CKD $1600 [57]
**PET**	Highly accurate with quantitative measurements of rest and stress myocardial blood flow, absolute and relative flow reserve [13]	Quantitative PET measurements especially flow reserve and myocardial blood flow are strongly associated with adverse patient outcomes such as cardiac mortality, with a greater prognostic value than SPECT [13]	Quantitative measurements permit better recognition of focal epicardial and diffuse microvascular disease	Limited availability	Radiation exposure: 2–5 mSv [57] Safe in CKD $1800 [57]
**CMR**	Dobutamine stress CMR in transplant candidates has been reported to have a sensitivity of 100%, a specificity of 89% for detecting angiographically significant CAD [60]	Due to fears about gadolinium-based contrast agents, the prognostic value of stress MRI perfusion studies has not been tested in CKD patients	Permits assessment of cardiomyopathy, cardiac remodelling, infarction, myocardial fibrosis and myocardial infiltration	Vasodilator stress CMR perfusion studies require gadolinium-based contrast agents Limited availability	No radiation exposure Gadolinium-based contrast agents pose a risk of nephrogenic systemic fibrosis [13]. Newer macrocyclic gadolinium-based contrast agents are substantially safer [66] $3700 [57]
**CACS**	High sensitivity but limited specificity in CKD patients who have vascular wall medial calcifications	Winther et al. [7] reported that a CACS > 400 has a greater prognostic value of MACE compared to risk factors and SPECT and is equivalent to CCTA and ICA	The total volume of CAC is a surrogate for plaque burden and CAD High negative predictive value	Statins increase CACS Localization of CAC does not correlate to vulnerable plaques.	$1600 [57] Radiation exposure: 1 mSV [57]
**CCTA**	High sensitivity (in the 90s) but limited specificity in CKD patients who have extensive coronary artery calcium	CCTA is a strong predictor of MACE, morbidity and mortality [7,13]	Can assess the degree of coronary stenosis, plaque volume, plaque characteristics, plaque vulnerability and when combined with perfusion or FFR, functional severity of stenosis can be assessed [57] High negative predictive value Preferred test in patients with a lower range of clinical likelihood of CAD [52]	Heart rate should be slow (<65 beats per minute) and regular Brief breath-holding is required to minimize motion artefact Atrial fibrillation (AF) is a relative contraindication Stenosis identified is not necessarily functionally significant, follow-on functional testing is recommended to evaluate the ischemic significance	Radiation exposure: 3–10 mSv [57] Risk of contrast-induced nephropathy from iodinated contrast agents $1600 [57]
**ICA**	Although previously considered the gold standard for the diagnosis of CAD, with the growth in non-invasive imaging modalities, it is reserved for patients whose clinical risk is high or when stress testing indicates significant ischemic burden [64]	Winther et al. [7] found that obstructive stenosis at ICA was associated with MACE but not mortality	Permits functional evaluation and hemodynamic assessment of stenosis Guides revascularisation options and permits simultaneous revascularisation when indicated	It is a lumenogram not an arteriogram: reduced sensitivity for diffuse disease and eccentric disease Risk of bleeding requiring blood transfusions is approximately 0.5–2% [52] Composite rate of death, MI or stroke is 0.1–0.2% [52]	7–9 mSv [57] Risk of contrast nephropathy should be reserved for patients with a high risk for CAD and those who would benefit from revascularisation

AF = atrial fibrillation; AVF = arteriovenous fistulas; CACS = coronary artery calcium score; CAD = coronary artery disease; CCTA = coronary computed tomography angiography; CKD = chronic kidney disease; CMR = cardiac magnetic resonance; DSE = dobutamine stress echocardiography; FFR = fractional flow reserve; GBCA = gadolinium-based contrast agents; ICA = invasive coronary angiography; LV = left ventricular; MACE = major adverse cardiac events; PET = positron emission tomography; SPECT = single photon emission computed tomography.

**Table 5 jcm-10-02525-t005:** Optimal medical therapy for CAD in patients with CKD [52].

Aspirin 75 mg daily in all patients Maximum tolerated dose of statin: the benefits of statin-based treatment become smaller as eGFR declines, with no evidence among patients on dialysis. If the goals are not achieved with the maximum tolerated dose of a statin, the addition of ezetimibe is recommended. For patients at very high risk who do not achieve their goal on a maximum tolerated dose of statin and ezetimibe, a combination with a PCSK9 inhibitor is recommended. Angiotensin-converting enzyme inhibitors (ACEi) or Angiotensin receptor blockers (ARB) are recommended in patients with HF, hypertension or diabetes Beta-blockers are recommended in patients with LV dysfunction or systolic HF

**Table 6 jcm-10-02525-t006:** Indications for revascularisation in patients with stable CAD to improve prognosis [76].

Left main disease with stenosis > 50% (Class I A ^1^) Proximal LAD stenosis > 50% (Class I A) Two-or-three-vessel disease with stenosis > 50% with impaired LV function (EF ≤ 35%) (Class I A) Large area of ischemia detected by functional testing (>10% LV) or abnormal invasive fractional flow reserve (Class I B ^2^) A single remaining patent coronary artery with stenosis > 50%

CAD = coronary artery disease; LAD = left anterior descending artery; EF = ejection fraction. ^1^ Class I A level of recommendation- highly recommended based on evidence from more than 1 high-quality RCT (randomised control trial); ^2^ Class 1 B level of recommendation- highly recommended based on evidence from more than 1 moderate-quality RCT.

**Table 7 jcm-10-02525-t007:** Decision-making between PCI and CABG [76].

Favours PCI	Favours CABG
Clinical characteristics: severe co-morbidities, advanced age, frailty, reduced life expectancy, restricted mobility Anatomical aspects: SYNTAX score 0–22, anatomy likely resulting in incomplete revascularisation with CABG due to poor quality or missing conduits Technical aspects: severe chest deformation or scoliosis	Clinical characteristics: diabetes, impaired LV function (EF < 35%), contraindication to DAPT, recurrent in-stent restenosis Anatomical aspects: SYNTAX score ≥ 23, anatomy likely resulting in incomplete revascularisation with PCI Technical aspects: severely calcified lesions limiting lesion expansion

CABG = coronary artery bypass grafting; DAPT = dual antiplatelet therapy; EF = ejection fraction; LV = left ventricular; PCI = percutaneous coronary intervention; SYNTAX = Synergy between PCI with TAXYS abd cardiac surgery.

**Table 8 jcm-10-02525-t008:** The use of oral anticoagulants in advanced CKD (CrCl < 30 mL/min/1.73 m^2^).

Oral Anticoagulant	Evidence	Recommendations [86]
Warfarin	A subgroup analysis from the SPAF (Stroke Prevention in Atrial Fibrillation) III trials showed that the efficacy of warfarin was broadly similar in stage 3 CKD patients and patients without CKD [87]	At all levels of kidney function, maintain time in therapeutic range ≥ 70% [93,94]
Dabigatran 80% renal excretion	RE-LY trial [99] excluded patients with CrCl < 30 mL/min/1.73 m^2^	In the USA only, CrCl 15–29 mL/min/1.73 m^2^: 75 mg Other areas, CrCl < 30 mL/min/1.73 m^2^: Do not use
Rivaroxaban 33% renal excretion	ROCKET-AF trial [100] excluded patients with CrCl < 30 mL/min/1.73 m^2^	CrCl 15–29 mL/min/1.73 m^2^: 15 mg once a day CrCl < 15 mL/min/1.73 m^2^: Do not use
Apixaban 27% renal excretion	ARISTOTLE trial [101] excluded patients with CrCl < 25 mL/min/1.73 m^2^ Lower risk of major bleeding events with apixaban than with warfarin in patients with CKD	CrCl 15–29 mL/min/1.73 m^2^: 2.5 mg twice daily In the USA only, CrCl < 15 mL/min/1.73 m^2^ or stable ESKD on dialysis: 5 mg twice daily Other areas, CrCl < 15 mL/min/1.73 m^2^: Do not use
Edoxaban 50% renal excretion	ENGAGE-AF TIMI 48 trial [102] excluded patients with CrCl < 30 mL/min/1.73 m^2^	CrCl 15–29 mL/min/1.73 m^2^: 30 mg once daily CrCl < 15 mL/min/1.73 m^2^: Do not use

**Table 9 jcm-10-02525-t009:** Summary of main investigations and management strategies to consider for cardiovascular screening of renal transplant candidates.

Element Name	Explanation
Smoking counselling	High prevalence of smokers on the waiting list that can be successfully targeted to improve postoperative outcomes
Exercise program	A regular exercise program to increase cardiopulmonary reserve improving both surgical outcomes and cardiovascular health
ECG	Basic screening test to look for rhythm abnormalities, ischaemic changes and chamber hypertrophy
Arrhythmia management	Consideration of anticoagulation with either warfarin or DOACs with warfarin preferred in advanced CKD
TTE	Assess ventricular function, valvular function and PASP
Non-invasive functional cardiac imaging	Stress-induced imaging to identify perfusion abnormalities and ischemia
Non-invasive anatomical cardiac imaging	CTCA and cardiac MR to allow for better visualisation of coronary anatomy and identification of patient who may benefit from invasive coronary angiography
Coronary angiogram and revascularisation	Invasive imaging in high-risk patients to view coronary anatomy to guide decisions about appropriate revascularisation strategy (PCI or CABG) in patients meeting criteria
RHC	Measuring PAP, PCWP, PVR and cardiac output to manage any PAH as appropriate

## Data Availability

Statement not applicable as this review did not report new data.

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
