# Peer review of "Preoperative Cardiovascular Assessment of the Renal Transplant Recipient: A Narrative Review"

_jcm, 2021, doi:10.3390/jcm10112525_

Round 1
Reviewer 1 Report
The authors are commended for tackling this topical issue. Cardiac "screening" prior to kidney transplantation is a challenging topic, with a huge variation in clinical practice. In its essence, screening should "identify a serious condition at an early/asymptomatic stage to allow intervention which improves outcomes". The major challenge is that screening for CVD has not been demonstrated to improve outcomes in the context of kidney transplantation, and may actually prove detrimental by delaying transplantation. The authors allude to this, but there should be more discussion of this challenge.
A significant limitation is the absence of discussion of functional assessments for potential recipients eg. DASI scores, 6 minute walk tests. These should be discussed in the manuscript.
The suggested algorithm in Figure 1 does not seem to be supported by the text e.g. the paragraph on CACS clearly discusses its major limitations in patients with advanced CKD. Also, the issue with absence of evidence for improved outcomes is pertinent when proposing a complex and potentially costly testing algorithm.
Minor comments
P1, Line 30-32: these percentages for recipient deaths combined exceed 100%.
P3, line 102-106: the current KDIGO guidelines for recipient work up do not suggest echo for all. This is misleading. The referenced guidelines are also unclear on this.
P7, line 260-3: the CARP and DECREASE trials should be referenced
P11, Fig 1: the acronym CCTA reads CTCA
References: these should be reviewed as full citations are not provided for all journals referenced
Author Response
Dear Reviewer,
We appreciate the time and effort that you have dedicated to providing your valuable feedback on our manuscript. We are grateful to the reviewers for their insightful comments on our paper, their inputs have definitely helped to improve the manuscript.
We thank you for the opportunity to address the comments from the reviewers. We have amended the manuscript after taking on board the feedback. We have highlighted the changes within the manuscript and we hope that the reviewers are satisfied with these changes to the manuscript.
Here is a point-by-point response to your major concerns:
- The authors are commended for tackling this topical issue. Cardiac "screening" prior to kidney transplantation is a challenging topic, with a huge variation in clinical practice. In its essence, screening should "identify a serious condition at an early/asymptomatic stage to allow intervention which improves outcomes". The major challenge is that screening for CVD has not been demonstrated to improve outcomes in the context of kidney transplantation, and may actually prove detrimental by delaying transplantation. The authors allude to this, but there should be more discussion of this challenge.
We thank the reviewer for raising this important point. We have included a paragraph expounding on this important limitation of cardiovascular screening in kidney transplant patients.
Lines 64-77: Whether screening improves transplant outcomes or survival is uncertain and there is risk that screening can lead to harm, unnecessarily subject candidates to invasive procedures and delay or exclude patients from transplantation 17–19. There is also no evidence that pre-emptive coronary revascularization improves outcomes in asymptomatic patients with stable CAD20. De Lima et al21 in their retrospective study of 1,696 kidney transplant candidates found that screening led a significant number of non-invasive and invasive tests, that the identification of CAD was predictive of post-transplantation coronary events but not mortality and found that intervention did not alter survival. Dunn et al22 in their propensity score-matched cohort analysis of 17,304 kidney transplant recipients found that routine cardiac stress testing was not associated with a difference in rates of death and myocardial infarction within 30 days of transplantation. As a result, the subject of pre-transplant cardiac screening is controversial with some calling for it to be abolished 23. The argument is that negative results can provide false reassurance and positive results can lead to further, sometimes invasive, investigations and the associated costs, resources, delays, risks of radiation24 and intervention25.
Lines 328 – 346: In summary, screening for occult CAD in asymptomatic kidney transplant candidates is complex. It is important to acknowledge the limitations of various screening approaches and the lack of evidence that pre-transplantation screening for CAD in asymptomatic patients with CKD improves outcomes. However, as noted, screening can guide prognostication and candidacy for transplantation. There are still many unanswered questions in CAD screening in kidney transplant candidates such as the frequency of CAD screening while on the transplant waiting list, a question currently being investigated by the CARSK (Canadian-Australasian Randomised Trial for Screening Kidney) trial66 and whether CAD screening leads to improved outcomes. Theoretically, careful risk stratification, identification of low-risk candidates for transplantation and closer follow-up to ensure medication adherence can possibly lead to better outcomes. However, prospective randomized trials are needed to define which subsets of patients might benefit from pretransplant intensified medical management or from revascularisation.
In terms of evaluating the screening program, the WHO principles of screening advise that a screening program should have scientific evidence of effectiveness: currently, there is no strong evidence that cardiac screening in kidney transplant candidates improves outcomes; the screening program has mechanisms to minimize potential risks: currently not supported by discrepancies between clinical guidelines (Table 1) and that overall benefits of screening outweigh harm, which is yet to be proven in this setting 23.
- A significant limitation is the absence of discussion of functional assessments for potential recipients eg. DASI scores, 6 minute walk tests. These should be discussed in the manuscript.
We thank the reviewers for raising this pertinent point. We acknowledge the important role of functional assessment in cardiac screening in the pre-transplant setting and have added the following lines and table to reflect this.
Lines 152-170: Pre-transplant poor physical function and low physical activity36 is associated with worse outcomes during and after transplantation 37. A cohort study of 540 patients found an association between low physical activity and increased risk cardiovascular and all-cause mortality in kidney transplant recipients 38. Rosas et al in their prospective cohort study of 507 kidney transplant recipients, found that physical activity at the time of kidney transplantation is a strong predictor of all-cause mortality 36. There is also growing evidence that exercise training can benefit kidney transplant recipients39,40 .
However, in clinical practice and in studies on physical activity in kidney transplant candidates, there is not a standardised approach to functional status assessment 37. There is also a lack of consensus on management of poor functional reserve and at what point risk of transplantation outweighs benefits.
The ideal functional status assessment tool is one that evaluates several aspects of physical functioning, guides risk stratification and predicts outcomes. Assessment tools should be objective, easy to administer and reproducible. Today there are more than 75 functional status assessment tools, some of the most frequently used tools that have an evidence base in the transplant setting are discussed in Table 3.
Table 3: Functional status assessment tools that can be used to evaluate kidney transplant candidates |
|||
Assessment tool |
Examples and details |
Advantages |
Disadvantages |
Self-reported physical assessment questionnaires |
Short Form-36 physical function scale, Instrumental Activities of Daily Living, Duke Activity status index (DASI), Physical Activity Scale for the Elderly (PASE) |
Easy to conduct |
Subjective, inaccurate reporting, difficult to use them longitudinally to quantify improvement
Studies have found the questionnaires scores to be associated with outcomes such as mortality in CKD patients36,41,42 |
Physical performance measures |
Grip strength and 6-min walk test (6MWT) |
Easy to use, low or no cost, time efficiency
Objective
In small ESRD cohort studies, better performance on the 6MWT correlated with improved quality of life43 |
Assesses only specific functions and muscle groups
Grip strength is significantly worse in the arm with arteriovenous fistula and older ESRD patients are already known to have lower grip strength. 44
6MWT can be unreliable due to variability resulting from changes in volume status and timing around dialysis (slow 6MWT if fluid overloaded) 45 |
Short Performance physical battery |
Combines use of 3 physical, lower-extremity performance measures: standing balance, walking speed, and chair stand tests |
5-10 minutes to conduct
Objective
In a prospective study including 700 kidney transplant patients, the SPPB score was associated with post-transplantation mortality 46 |
Cannot be utilised in those with lower extremity abnormalities e.g. lower extremity amputations |
Fried’s Frailty Phenotype Score |
5 domains: weight loss, exhaustion, physical activity, grip strength, and walking speed |
Measured frailty by FFP scoring has shown correlation with post-transplantation outcomes 47,48 |
Unintentional weight loss and exhaustion—are subjective and self-reported
American Society of Transplantation frailty assessment survey results show that the FFP score was utilized by only 3.6% of the survey takers who reported assessing frailty for candidacy evaluation49 |
- The suggested algorithm in Figure 1 does not seem to be supported by the text e.g. the paragraph on CACS clearly discusses its major limitations in patients with advanced CKD. Also, the issue with absence of evidence for improved outcomes is pertinent when proposing a complex and potentially costly testing algorithm.
We accept that the proposed screening pathway can be confusing and perceived conflicting to the rest of the text so has been removed. We acknowledge that in the face of uncertainty, new evidence, new technologies and changing guidelines, decisions will be made on an individual level guided by patient, provider and center specific factors. The manuscript will serve as a guide for clinicians to make decisions about appropriate screening tests and the advantages and limitations of each approach.
- P3, line 102-106: the current KDIGO guidelines for recipient work up do not suggest echo for all. This is misleading. The referenced guidelines are also unclear on this.
We acknowledge this line can be confusing and that there are significant differences between guidelines. We have therefore removed the above lines and have added the following table containing a summary of the main guidelines.
Table 1: Summary of main recommendations from International Guidelines |
The 2020 Kidney Disease: Improving Global Outcomes (KDIGO) Clinical Practice Guideline12 · Evaluate all candidates for the presence and severity of cardiac disease with history, physical examination, and ECG · Candidates with signs or symptoms of cardiac disease should be referred to a cardiologist and undergo management before being considered for transplantation · Candidates at high risk for coronary artery disease (CAD) or with poor functional capacity should undergo noninvasive CAD screening · Asymptomatic candidates with known CAD should not be revascularized exclusively to reduce perioperative cardiac events · Patients with asymptomatic, advanced triple vessel coronary disease; uncorrectable, symptomatic New York Heart Association (NYHA) Functional Class III/IV heart disease should be excluded from kidney transplantation unless they have an estimated survival which is acceptable according to national standards · Asymptomatic candidates who have been on dialysis for at least two years or have risk factors for pulmonary hypertension should undergo echocardiography · Patients with an estimated pulmonary systolic pressure greater than 45 mm Hg, severe valvular heart disease, myocardial infarction should be assessed by a cardiologist
American Society of Transplantation (AST)13 (2002) · ECG and chest radiographer in all candidates · Echocardiogram if left ventricular hypertrophy, congestive heart failure or myocardial dysfunction suspected · Noninvasive stress testing recommended for patients at “high risk” (diabetes, known ischemic heart disease (IHD), or 2+ risk factors: age ≥ 45 in men or ≥55 in women, IHD in first degree relative, smoker, diabetes, hypertension, dyslipidemia, left ventricular hypertrophy) · Coronary angiography for patients with a positive stress test · Revascularization before transplantation for patients with critical coronary lesions
American College of Cardiology/American Heart Association (ACC/AHA)14 (2007) · Consider further cardiac evaluation in symptomatic patients · Does not encourage further testing for patients who have no cardiac symptoms with a functional capacity of 4+ metabolic equivalents of tasks (METs) regardless of diabetes, history of CAD, or other traditional cardiac risk factors · Consider noninvasive testing in asymptomatic patients with 1 or 2 clinical risk markers (ischemic heart disease, compensated or prior heart failure, diabetes, decreased kidney function, cerebrovascular disease) and poor functional capacity who require intermediate-risk noncardiac surgery if it will change management · Recommendations for testing are stronger if 3+ clinical risk factors are present
American Heart Association/American College of Cardiology Foundation (AHA/ACCF)15 (2012) • ECG and echocardiogram in all patients • Noninvasive stress testing in kidney transplant candidates with no active cardiac conditions based on the presence of ≥ CAD risk factors (diabetes, prior CVD, >1 y on dialysis, left ventricular hypertrophy, age > 60 y, smoking, hypertension, and dyslipidemia) regardless of functional status • Coronary angiography in patients who meet the criteria based on 2011 ACCF and AHA Guidelines for coronary artery bypass graft surgery
European Renal Best Practice (ERBP) 16 (2015) • In asymptomatic low-risk candidates, basic clinical data, physical examination, ECG at rest, and chest x-ray are a sufficient standard workup in • In asymptomatic high-risk patients, (older age, diabetes, personal or family history of CVD), standard exercise tolerance test and echocardiogram is recommended; in those with a true negative test result, further cardiac screening not indicated • In candidates with high risk and a positive or inconclusive exercise tolerance test, further cardiac investigation for occult CAD with noninvasive stress imaging (myocardial perfusion or dobutamine stress echocardiography) is recommended • Recommend coronary angiography in candidates with a positive test for cardiac ischemia |
- References: these should be reviewed as full citations are not provided for all journals referenced
The references have been corrected and full citations for all references are provided.
We once again thank the reviewer for their insightful comments and time in thoroughly reviewing our paper.
Reviewer 2 Report
In this review, the authors synthesize the assessment, the screening and management of coronary artery disease in kidney transplant candidates.
This review is well written, exhaustive and may be useful for centers that wish to implement cardiac monitoring in kidney transplant patients and candidates. Yet, there is already some review on this topic such as this one : PMID: 25296093
My comments:
- Summary of main recommendations are lacking (KDOQI, AST, ACC/AHA..)
- Page 3 “ AVF [..] can affect response to vasodilators”. Please add a reference
- The authors may also discuss, if data are available, the impact of the initial nephropathy on the CAD risk assessment: diabetes mellitus, hypertension, vasculitis
- Can authors be more precise regarding to the risk prediction scores (paragraph 5) maybe by giving a table or by giving some values on risks. Notably, the impact of proteinuria is of major relevance.
- This paper is important : PMID: 21511835 “A call to action: variability in guidelines for cardiac evaluation before renal transplantation” and should be discuss.
- Page 7 and 8 : authors should discuss the use of angiotensin converting enzyme inhibitors and angiotensin receptor blockers on late stage (4-5) of chronic renal failure. Those articles may be included in the discussion: PMID: 32150237
- Page 12, paragraph 18 about arrhythmias: the practical difficulty is to manage stage 4 and 5 with atrial fibrillation and the use of anticoagulation. The paragraph should focus on this subject and describe more precisely data on the different anticoagulants.
- Page 13, paragraph 20 about obesity: the authors should discuss more in depth the bariatric surgery. Should all obese kidney transplant candidates refer to bariatric surgery before or after transplantation? Does this strategy impact the tile on waiting list and cardiovascular morbi-mortality?
- Page 13, paragraph valvulopathies: should be also either more detailed about the different vasculopathies and therapeutic options.
- Page 14, paragraph pulmonary hypertension: in my opinion, the WHO classification is too detailed, which is not the purpose of the review. This may be replaced by a pathway of management and investigation to sum up how to proceed.
Minors comments:
- Page 4, line 161: please define ICA (further defined page 4 line 172).
- Page 4, CCTA is defined twice line 167 and 171
Author Response
Dear Reviewer,
We appreciate the time and effort that you have dedicated to providing your valuable feedback on our manuscript. We are grateful to the reviewers for their insightful comments on our paper, their inputs have definitely helped to improve the manuscript.
We thank you for the opportunity to address the comments from the reviewers. We have amended the manuscript after taking on board the feedback. We have highlighted the changes within the manuscript and we hope that the reviewers are satisfied with these changes to the manuscript.
Here is a point-by-point response to your specific concerns:
- Summary of main recommendations are lacking (KDOQI, AST, ACC/AHA..)
The authors thank the reviewer for highlighting this point and the authors agree that a summary of the main recommendations are necessary. We have therefore added the following lines and table to the manuscript.
Lines 58-60: Screening for occult CAD among kidney transplant candidates is recommended by guidelines to identify candidates with occult disease, however there are significant disparities between the different guidelines (Table 1).
A table containing a summary of the main recommendations has been added
Table 1: Summary of main recommendations from International Guidelines |
The 2020 Kidney Disease: Improving Global Outcomes (KDIGO) Clinical Practice Guideline12 · Evaluate all candidates for the presence and severity of cardiac disease with history, physical examination, and ECG · Candidates with signs or symptoms of cardiac disease should be referred to a cardiologist and undergo management before being considered for transplantation · Candidates at high risk for coronary artery disease (CAD) or with poor functional capacity should undergo noninvasive CAD screening · Asymptomatic candidates with known CAD should not be revascularized exclusively to reduce perioperative cardiac events · Patients with asymptomatic, advanced triple vessel coronary disease; uncorrectable, symptomatic New York Heart Association (NYHA) Functional Class III/IV heart disease should be excluded from kidney transplantation unless they have an estimated survival which is acceptable according to national standards · Asymptomatic candidates who have been on dialysis for at least two years or have risk factors for pulmonary hypertension should undergo echocardiography · Patients with an estimated pulmonary systolic pressure greater than 45 mm Hg, severe valvular heart disease, myocardial infarction should be assessed by a cardiologist
American Society of Transplantation (AST)13 (2002) · ECG and chest radiographer in all candidates · Echocardiogram if left ventricular hypertrophy, congestive heart failure or myocardial dysfunction suspected · Noninvasive stress testing recommended for patients at “high risk” (diabetes, known ischemic heart disease (IHD), or 2+ risk factors: age ≥ 45 in men or ≥55 in women, IHD in first degree relative, smoker, diabetes, hypertension, dyslipidemia, left ventricular hypertrophy) · Coronary angiography for patients with a positive stress test · Revascularization before transplantation for patients with critical coronary lesions
American College of Cardiology/American Heart Association (ACC/AHA)14 (2007) · Consider further cardiac evaluation in symptomatic patients · Does not encourage further testing for patients who have no cardiac symptoms with a functional capacity of 4+ metabolic equivalents of tasks (METs) regardless of diabetes, history of CAD, or other traditional cardiac risk factors · Consider noninvasive testing in asymptomatic patients with 1 or 2 clinical risk markers (ischemic heart disease, compensated or prior heart failure, diabetes, decreased kidney function, cerebrovascular disease) and poor functional capacity who require intermediate-risk noncardiac surgery if it will change management · Recommendations for testing are stronger if 3+ clinical risk factors are present
American Heart Association/American College of Cardiology Foundation (AHA/ACCF)15 (2012) • ECG and echocardiogram in all patients • Noninvasive stress testing in kidney transplant candidates with no active cardiac conditions based on the presence of ≥ CAD risk factors (diabetes, prior CVD, >1 y on dialysis, left ventricular hypertrophy, age > 60 y, smoking, hypertension, and dyslipidemia) regardless of functional status • Coronary angiography in patients who meet the criteria based on 2011 ACCF and AHA Guidelines for coronary artery bypass graft surgery
European Renal Best Practice (ERBP) 16 (2015) • In asymptomatic low-risk candidates, basic clinical data, physical examination, ECG at rest, and chest x-ray are a sufficient standard workup in • In asymptomatic high-risk patients, (older age, diabetes, personal or family history of CVD), standard exercise tolerance test and echocardiogram is recommended; in those with a true negative test result, further cardiac screening not indicated • In candidates with high risk and a positive or inconclusive exercise tolerance test, further cardiac investigation for occult CAD with noninvasive stress imaging (myocardial perfusion or dobutamine stress echocardiography) is recommended • Recommend coronary angiography in candidates with a positive test for cardiac ischemia |
- Page 3 “ AVF [..] can affect response to vasodilators”. Please add a reference
Thank you highlighting this. We have added the relevant references.
Line 202: arteriovenous fistulas (AVFs) used for haemodialysis can affect response to vasodilators 56,57
- The authors may also discuss, if data are available, the impact of the initial nephropathy on the CAD risk assessment: diabetes mellitus, hypertension, vasculitis
The authors agree that a discussion on the impact of nephropathy on CAD risk assessment is warranted. We have therefore added the following lines to the manuscript to reflect this important concept.
Lines 48 – 56: Many CKD patients remain asymptomatic despite developing severe CAD and there is a high prevalence of silent myocardial ischemia owing to factors such as diabetic and uremic neuropathy in CKD patients 7,8. Benett et al9 examined 11 asymptomatic diabetic ESRD patients who voluntarily underwent coronary angiography and found multivessel CAD in all patients. Weinrauch et al10 examined 21 ESRD with Type 1 diabetes with no clinical or ECG evidence of CAD and found that 50% had CAD and 38% had significant CAD. Dyspnea on exertion is also less specific for angina as it may be secondary to anemia, volume overload, or metabolic acidosis in patients with CKD 7. CAD amongst patients with CKD is also not universal: as many as 50% to 70% of patients with advanced CKD do not have obstructive CAD 11.
- Can authors be more precise regarding to the risk prediction scores (paragraph 5) maybe by giving a table or by giving some values on risks.
The authors thank the reviewer for highlighting this. We have amened the manuscript and discuss further about risk prediction score.
Lines 112-119: Risk prediction scores such as Framingham risk score (FRS) commonly underestimate risk of CAD in CKD patients 27. Silver et al27 in their retrospective study of 956 kidney transplant recipients found that FRS substantially underestimated MACE (actual-to-predicted event ratio 1.2-8.4 in different subgroups, all P<0.0001) and found in their multivariate COX modeling that only FRS ≥ 10% and eGFR predicted MACE. The addition of CRP, uric acid and urine albumin-to-creatinine ratio were not found to increase the prediction of MACE. The greatest underestimation of risk occurred in patients with preexisting IHD, diabetes and smoking history. A number of other composite risk scores have been developed, but few have been externally validated28.
- Notably, the impact of proteinuria is of major relevance.
We acknowledge the relevance of proteinuria and thank the reviewer for highlighting this pertinent point. We have added the following lines to the manuscript.
Lines 132-137: Studies have found proteinuria to be predictive for CVD and associated with mortality and morbidity 32. In one study, a higher urinary albumin concentration increased the risk of cardiovascular death after adjusting for other cardiovascular risk factors 33. Bello et al demonstrated that proteinuria at each stage of CKD was associated with a higher risk of CVD 34. These studies suggest a role for proteinuria in the pre-transplant setting to risk-stratify patients and identify those at an increased for CVD.
- This paper is important : PMID: 21511835 “A call to action: variability in guidelines for cardiac evaluation before renal transplantation” and should be
We agree with the reviewers that the topic of cardiac screening is rife with uncertainty and is continuously evolving as more evidence and new technologies allow more accurate assessment and better understanding of screening results. This results in great variability in guidelines as new findings are incorporated.
Lines 58-77: Screening for occult CAD among kidney transplant candidates is recommended by guidelines to identify candidates with occult disease, however there are significant disparities between the different guidelines (Table 1).
Whether screening improves transplant outcomes or survival is uncertain and there is risk that screening can lead to harm, unnecessarily subject candidates to invasive procedures and delay or exclude patients from transplantation 17–19. There is also no evidence that pre-emptive coronary revascularization improves outcomes in asymptomatic patients with stable CAD20. De Lima et al21 in their retrospective study of 1,696 kidney transplant candidates found that screening led a significant number of non-invasive and invasive tests, that the identification of CAD was predictive of post-transplantation coronary events but not mortality and found that intervention did not alter survival. Dunn et al22 in their propensity score-matched cohort analysis of 17,304 kidney transplant recipients found that routine cardiac stress testing was not associated with a difference in rates of death and myocardial infarction within 30 days of transplantation. As a result, the subject of pre-transplant cardiac screening is controversial with some calling for it to be abolished 23. The argument is that negative results can provide false reassurance and positive results can lead to further, sometimes invasive, investigations and the associated costs, resources, delays, risks of radiation24 and intervention25.
- Page 7 and 8 : authors should discuss the use of angiotensin converting enzyme inhibitors and angiotensin receptor blockers on late stage (4-5) of chronic renal failure. Those articles may be included in the discussion: PMID: 32150237
We thank the reviewer for this raising this important point. We acknowledge its value and have added the following lines to the manuscript.
Lines 382 – 386: Guidelines recommend continuing maintenance cardioprotective medications including beta-blockers, statins, aspirin and angiotensin-converting enzyme inhibitors while waiting for kidney transplantation and also in the perioperative period. Qiao et al 73 found that discontinuing ACE-I or ARB therapy in patients with declining kidney function was associated with a higher risk of mortality, MACE but no statistically significant difference in the risk of ESKD.
- Page 12, paragraph 18 about arrhythmias: the practical difficulty is to manage stage 4 and 5 with atrial fibrillation and the use of anticoagulation. The paragraph should focus on this subject and describe more precisely data on the different anticoagulants.
We thank the reviewer for raising this pertinent concern. We acknowledge the difficulties encountered in making decisions about anticoagulation strategy especially in those with advanced CKD, as will be the case in kidney transplant candidates, and have added the following lines to reflect this.
Lines 458 – 486: Evidence from RCTs support the safe use of warfarin and DOACs in CKD stages 1 to 3: DOACs, with superior safety profile and lower bleeding risk, are preferred in CKD stages 1 to 3 88. However, evidence is sparse and conflicting in more advanced stages of CKD (CrCl <25-30 ml/min), ESRD and dialysis with these patient groups being excluded from the large RCTs investigating the efficacy of warfarin and DOACs 88,89.
A subgroup analysis from the SPAF (Stroke prevention in Atrial Fibrillation) III trials showed that the efficacy of warfarin was broadly similar in stage 3 CKD patients and patients without CKD 90. However, caution is warranted on the use of warfarin in patients with more advanced CKD: a meta-analysis of 13 studies found that warfarin use in patients with ESRD had a neutral effect on risk of ischemic stroke and all-cause mortality and was associated with a significantly increased risk of major bleeding91. Other studies also report similar neutral effects of warfarin on the risk of ischemic stroke and thromboembolic events, with some even reporting increased risk of ischemic stroke92–94. A Swedish nationwide cohort study showed that patients with AF and CKD or ESRD would benefit from warfarin if stroke and bleeding risk factors are optimally managed and there is tight control on anticoagulation95. In CKD patient receiving warfarin, a time in the therapeutic range (TTR) of >70% independently predicted reduced risk of stroke, death and major bleeding but the risk of suboptimal TTR (<65%) increased in the presence of CKD96,97.
With the lack of evidence from RCTs on the efficacy of DOACs in advanced CKD, findings from pharmacological modelling have been used to guide practice. In Europe, reduced doses of rivaroxaban, apixaban and edoxaban have been approved to be used in patients with severe CKD (CrCl 15-29 ml/min) not on dialysis (Table 8)98,99 with the European Society of Cardiology AF guidelines emphasizing that there are no randomized controlled trials on the use of DOACs in patients with severe CKD99. The US Food and Drug administration also approved dabigatran in patients with CrCl 15-29 ml/min and the use of apixaban in patients with stable ESRD on dialysis89. The available data highlight that correct dosing of DOACs is essential: in a large cohort study, underdosing or overdosing of DOACs was associated with decreased safety100. Assessment of renal function before starting a DOAC and regular monitoring is advised.
Table 8: The use of oral anticoagulants in advanced CKD (CrCl < 30 ml/min/1.73 m2) |
||
Oral Anticoagulant |
Evidence |
Recommendations89 |
Warfarin |
A subgroup analysis from the SPAF (Stroke prevention in Atrial Fibrillation) III trials showed that the efficacy of warfarin was broadly similar in stage 3 CKD patients and patients without CKD90 |
At all levels of kidney function, maintain time in therapeutic range ≥70%96,97 |
Dabigatran
80% renal excretion
|
RE-LY trial102 excluded patients with CrCl <30 ml/min/1.73 m2
|
In the USA only, CrCl 15-29 ml/min/1.73 m2: 75mg
Other areas, CrCl <30 ml/min/1.73 m2: Do not use |
Rivaroxaban
33% renal excretion
|
ROCKET-AF trial103 excluded patients with CrCl <30 ml/min/1.73 m2 |
CrCl 15 – 29 ml/min/1.73 m2: 15mg once a day
CrCl <15 ml/min/1.73 m2: Do not use |
Apixaban
27% renal excretion
|
ARISTOTLE trial104 excluded patients with CrCl <25 ml/min/1.73 m2
Lower risk of major bleeding events with apixaban than with warfarin in patients with CKD |
CrCl 15 - 29 ml/min/1.73 m2: 2.5mg twice daily
In the USA only, CrCl <15 ml/min/1.73 m2 or stable ESRD on dialysis: 5mg twice daily
Other areas, CrCl <15 ml/min/1.73 m2: Do not use |
Edoxaban
50% renal excretion
|
ENGAGE-AF TIMI 48 trial105 excluded patients with CrCl <30 ml/min/1.73 m2
|
CrCl 15 – 29 ml/min/1.73 m2: 30mg once daily
CrCl <15 ml/min/1.73 m2: Do not use |
- Page 13, paragraph 20 about obesity: the authors should discuss more in depth the bariatric surgery. Should all obese kidney transplant candidates refer to bariatric surgery before or after transplantation? Does this strategy impact the tile on waiting list and cardiovascular morbi-mortality?
Thank you for your comment. This is certainly a most interesting topic that has now been included in the section on obesity. The benefits of bariatric surgery are clear but more research is needed on the timing of bariatric surgery in these patients.
- Page 13, paragraph valvulopathies: should be also either more detailed about the different vasculopathies and therapeutic options.
We thank the reviewer for their comment and have included some extra detail on the treatment on aortic stenosis. We have also included reference to some guidelines on the treatment of valvular heart diseases. We would however argue that speicific treatment of valve disorders falls out of the scope of this review.
- Page 14, paragraph pulmonary hypertension: in my opinion, the WHO classification is too detailed, which is not the purpose of the review. This may be replaced by a pathway of management and investigation to sum up how to proceed.
We thank the reviewer for their comment. While we agree that there is significant detail on the WHO classificatoin of pulmonary hypertension, we believe that this information is not covered in other reviews on the cardiovascular optimisation of the renal transplant patient. We also argue that a lot of the causes of pulmonary hypertension are linked to renal pathologies as evidence by the high preavalence of this disease in potential renal transplant candidates. We have also included a pathway for renal transplant candidates; right heart catherisation in patients with elevated pulmonary artery systolic pressure on transthoracic echocardiogram with further management based on the type of pulmonary hypertesion.
We once again thank the reviewer for their insightful comments and time in thoroughly reviewing our paper.
Reviewer 3 Report
Dear Editor, Thank you for the opportunity to review the paper: " Preoperative Cardiovascular Assessment and Optimization of the Renal Transplant Recipient: A Review."
The presented publication aims to improve care and unify perioperative diagnosis in patients undergoing KTx. However, the paper adds nothing new beyond the published and practiced ERA-EDTA (European Renal Association-European Dialysis and Transplant Association) Therapy Standards 1, 2
The paper is written incoherently and vaguely, besides the basic known information about coronary stenosis causing impaired blood flow "...CAD is a narrowing or blockage of the arteries supplying the heart caused by atherosclerosis..." we find information on modern and not fully validated imaging modalities and functional tests to assess cardiac blood supply and function. [46-47] [107-209].
Also striking and incomprehensible to transplantation thought is a very superficial statement defining the indications for kidney transplantation and an entire monologue defining optimal recipients: "...With the scarcity of organs for transplantation and the imbalance between the availability of organs and the number of patients on the transplant waiting list, there is a necessity to identify candidates with the highest likelihood [37-42]. Information is omitted that these are often relative contraindications and therefore temporary.
The description of asymptomatic CAD in uremic and diabetic patients is also contrary to basic cardiology knowledge. Of course, diabetic and uremic autonomic neuropathy render patients asymptomatic with respect to typical CAD symptoms, but otherwise ischemic symptoms are reflected in ECG, troponin levels, and TTE. According to the authors:" ...Furthermore, many patients with CKD remain asymptomatic despite developing severe CAD because comorbidities such as diabetes and uremia often render them asymptomatic with respect to typical angina symptoms. Patients with CKD do not show typical clinical signs of CAD and do not exhibit the typical changes seen in CAD on the electrocardiogram (ECG), such as ST-T changes and abnormal Q waves [58-63]."
The paper is wordy, adds nothing new to the clinic of patients, and is similar to published and clearly described papers 3. It cannot be considered as the basis of a schematic approach for the transplant team preparing a patient for KTx.
The paper is not suitable for publication.
1. European Renal Best Practice Guideline on kidney donor and recipient evaluation and perioperative care, Nephrology Dialysis Transplantation, Volume 30, Issue 11, November 2015, Pages 1790-1797,
2. Renal replacement therapy in Europe: a summary of the 2013 Registry Annual Report with a focus on diabetes mellitus. Clin Kidney J. 2016 Jun;9(3):457-69. doi: 10.1093/ckj/sfv151. Epub 2016 Jan 31.
3. Pilmore H. Cardiac assessment for renal transplantation. Am J Transplant. 2006 Apr;6(4):659-65. doi: 10.1111/j.1600-6143.2006.01253.x.
Author Response
Dear Reviewer,
We appreciate the time and effort that you have dedicated to providing your valuable feedback on our manuscript. We are grateful to the reviewers for their insightful comments on our paper.
We have been able to incorporate changes to reflect most of the suggestions provided by you. We have highlighted the changes within the manuscript.
Here is a point-by-point response to your specific concerns:
Dear Editor, Thank you for the opportunity to review the paper: " Preoperative Cardiovascular Assessment and Optimization of the Renal Transplant Recipient: A Review."
The presented publication aims to improve care and unify perioperative diagnosis in patients undergoing KTx. However, the paper adds nothing new beyond the published and practiced ERA-EDTA (European Renal Association-European Dialysis and Transplant Association) Therapy Standards 1, 2
We thank the reviewer for their detailed review of our paper. While we do agree that there exist guidelines on the cardiovascular management of kidney transplant patients, our paper provides a detailed review exploring some of the evidence behind these recommendations. Our paper also highlights areas for future work including functional cardiovascular imaging, the role of bariatric surgery and the management of pulmonary hypertension. We also highlight data from the latest trials showing the controversies behind cardiovascular screening in this patient population.
The paper is written incoherently and vaguely, besides the basic known information about coronary stenosis causing impaired blood flow "...CAD is a narrowing or blockage of the arteries supplying the heart caused by atherosclerosis..." we find information on modern and not fully validated imaging modalities and functional tests to assess cardiac blood supply and function. [46-47] [107-209].
We thank the reviewer for their comments. We agree that some imaging modalities mentioned have not been specifically shown to improve outcomes in potential renal transplant candidates, we argue that echocardiography, stress echocardiography, computerised tomography coronary angiography, cardiac magnetic resonance imaging, single-photon emission computed tomography and positron emission tomography have been established, evidence-based methods for assessing myocardial function. We aim for this review to be a starting point for centres looking to investigate novel methods of assessing cardiac function in renal transplant candidates.
Also striking and incomprehensible to transplantation thought is a very superficial statement defining the indications for kidney transplantation and an entire monologue defining optimal recipients: "...With the scarcity of organs for transplantation and the imbalance between the availability of organs and the number of patients on the transplant waiting list, there is a necessity to identify candidates with the highest likelihood [37-42]. Information is omitted that these are often relative contraindications and therefore temporary.
We thank the reviewer for their comment. We have added further language to this sentence to highlight that cardiovascular screening can guide transplantation rather than dictate transplantation.
The description of asymptomatic CAD in uremic and diabetic patients is also contrary to basic cardiology knowledge. Of course, diabetic and uremic autonomic neuropathy render patients asymptomatic with respect to typical CAD symptoms, but otherwise ischemic symptoms are reflected in ECG, troponin levels, and TTE. According to the authors:" ...Furthermore, many patients with CKD remain asymptomatic despite developing severe CAD because comorbidities such as diabetes and uremia often render them asymptomatic with respect to typical angina symptoms. Patients with CKD do not show typical clinical signs of CAD and do not exhibit the typical changes seen in CAD on the electrocardiogram (ECG), such as ST-T changes and abnormal Q waves [58-63]."
We thank the reviewer for this comment. We have rewritten this paragraph with further references highlighting the challenge of diagnosing coronary artery disease. We do not wish to imply that features of coronary artery disease are completely different in patients with normal kidney function compared to a patient with renal disease. However, we include references to show that significant differences do exist in the presentation of CAD that clinicians must be aware of.
The paper is wordy, adds nothing new to the clinic of patients, and is similar to published and clearly described papers 3. It cannot be considered as the basis of a schematic approach for the transplant team preparing a patient for KTx.
The paper is not suitable for publication.
We accept that there are many good papers on the topic of cardiac assessment in renal transplantation, we believe our paper covers the latest trials and a broader range of topics. The paper by Pilmore et al. is an excellent example of coronary screening in end-stage renal disease but our paper also includes research on arrhythmias, valvular pathology, smoking, obesity, pulmonary hypertension and heart failure. We have also extensively rewritten the paper based on changes suggested by the reviewers.
We once again thank you for your comments and look forward to your further review.
Reviewer 4 Report
Major remarks:
- Surprisingly, the authors do not mention the controversy on routine CAD screening in waitlisted Tx-patients (see also 1053/j.ajkd.2019.05.033).
- Furthermore, I miss a paragraph on myocardial scintigraphy.
- Therefore, I would recommend a more balanced view with citations of positive and negative studies in this field having also in mind potential publication bias of reporting only significant results in this field.
- Does routine CAD screening fulfill epidemiological screening criteria?
- Blood test in CKD patients: there should be a commentary on the fact that troponin levels are commonly elevated in ESRD pts and that their evaluation over time should be monitored
- …end-stage renal disease (ESRD) – please provide a citation (introduction)
- The authors should point out more clearly what is new in their work as compared to former revies on this topic (see citation above also)
Check editing or spelling:
Patient awaiting renal transplant often…
among patients in the kidney….
Optimization – optimisation – please use American or English style only throughout the whole manuscript
…points towards underlying CAD.
Author Response
Dear Reviewer,
We appreciate the time and effort that you have dedicated to providing your valuable feedback on our manuscript. We are grateful to the reviewers for their insightful comments on our paper, their inputs have definitely helped to improve the manuscript.
We thank you for the opportunity to address the comments from the reviewers. We have amended the manuscript after taking on board the feedback. We have highlighted the changes within the manuscript and we hope that the reviewers are satisfied with these changes to the manuscript.
Here is a point-by-point response to your specific concerns:
- Surprisingly, the authors do not mention the controversy on routine CAD screening in waitlisted Tx-patients (see also 1053/j.ajkd.2019.05.033)
The authors acknowledge that the subject of routine CAD screening in the pre-transplantation setting is controversial. To highlight this, the following text has been added to the manuscript.
Lines 58 – 77: Screening for occult CAD among kidney transplant candidates is recommended by guidelines to identify candidates with occult disease, however there are significant disparities between the different guidelines (Table 1).
Whether screening improves transplant outcomes or survival is uncertain and there is risk that screening can lead to harm, unnecessarily subject candidates to invasive procedures and delay or exclude patients from transplantation 17–19. There is also no evidence that pre-emptive coronary revascularisation improves outcomes in asymptomatic patients with stable CAD20. De Lima et al21 in their retrospective study of 1,696 kidney transplant candidates found that screening led a significant number of non-invasive and invasive tests, that the identification of CAD was predictive of post-transplantation coronary events but not mortality and found that intervention did not alter survival. Dunn et al22 in their propensity score-matched cohort analysis of 17,304 kidney transplant recipients found that routine cardiac stress testing was not associated with a difference in rates of death and myocardial infarction within 30 days of transplantation. As a result, the subject of pre-transplant cardiac screening is controversial with some calling for it to be abolished 23. The argument is that negative results can provide false reassurance and positive results can lead to further, sometimes invasive, investigations and the associated costs, resources, delays, risks of radiation24 and intervention25.
- Furthermore, I miss a paragraph on myocardial scintigraphy.
The authors thank the reviewer for highlighting this point and we have amended the manuscript to discuss each type of functional imaging individually, making it easier to identify and read about a specific type of imaging modality. We have clarified that SPECT is a form of MPS.
Lines 229 – 242: Myocardial perfusion scintigraphy (MPS) can utilize SPECT, a nuclear imaging test which uses radioactive tracers to trace blood flow and cardiac perfusion. It can be utilised in patients with uncontrolled blood pressure or arrhythmias 51. Dipyridamole is the typical pharmacological stressor used and works by increasing adenosine levels, causing vasodilatation. The limitation in CKD patients is that the already higher basal adenosine levels attenuates detection of stressor induced perfusion abnormalities 26. In addition, the common utilisation of anti-anginal and antihypertensive medicines by CKD patients, as described above, reduces sensitivity further. Another limitation is that attenuation correction is required to correct artefacts, resulting in low image quality. There is also considerable radiation exposure but it is safe to use in CKD patients.
Prognostically, an abnormal SPECT scan nearly doubles the risk of death in CKD patients 4 and a normal SPECT is associated with a relatively low risk of future adverse events 51, 52 It is widely available and quantification of blood flow is now possible, overcoming limitations of subjective interpretation of flow abnormalities.
- Therefore, I would recommend a more balanced view with citations of positive and negative studies in this field having also in mind potential publication bias of reporting only significant results in this field.
The authors agree that a more balanced view on the subject of cardiac screening, citing both positive and negative tests, is required to give the reader a more realistic understanding of this field. Therefore, we have added the following lines and for each method of screening, we have discussed both the advantages and limitations.
Lines 64 – 88: Whether screening improves transplant outcomes or survival is uncertain and there is risk that screening can lead to harm, unnecessarily subject candidates to invasive procedures and delay or exclude patients from transplantation 17–19. There is also no evidence that pre-emptive coronary revascularisation improves outcomes in asymptomatic patients with stable CAD20. De Lima et al21 in their retrospective study of 1,696 kidney transplant candidates found that screening led a significant number of non-invasive and invasive tests, that the identification of CAD was predictive of post-transplantation coronary events but not mortality and found that intervention did not alter survival. Dunn et al22 in their propensity score-matched cohort analysis of 17,304 kidney transplant recipients found that routine cardiac stress testing was not associated with a difference in rates of death and myocardial infarction within 30 days of transplantation. As a result, the subject of pre-transplant cardiac screening is controversial with some calling for it to be abolished 23. The argument is that negative results can provide false reassurance and positive results can lead to further, sometimes invasive, investigations and the associated costs, resources, delays, risks of radiation24 and intervention25.
However, in the setting of transplantation, there is the argument that screening can guide decisions about transplant candidacy 4 and direct pretransplant optimisation. Screening can identify individuals with a high burden of CAD that confers poor prognosis and there is evidence, presented below, that screening tests can guide prognostication. Though, given the uncertain and controversial subject, the risk of publication bias towards studies that found significant results should be taken into account. Transplant candidates spend many months and years on the transplant waiting list and screening can be used monitor for development of CAD, initiate treatment and maintain medical fitness. Perioperative events can severely impact transplanted kidney function and with pre-operative optimisation, it is hoped (but not proven) that treatment invoked by screening may prevent perioperative events and improve long-term outcomes.
- Does routine CAD screening fulfill epidemiological screening criteria?
We acknowledge the importance of discussing cardiac screening in the context of broader epidemiological screening criteria as it will allow us to better judge the utility of screening and guide improvements in screening strategy. We have added the following lines to manuscript to reflect this.
Lines 341 – 346: In terms of evaluating the screening program, the WHO principles of screening advise that a screening program should have scientific evidence of effectiveness: currently, there is no strong evidence that cardiac screening in kidney transplant candidates improves outcomes; the screening program has mechanisms to minimize potential risks: currently not supported by discrepancies between clinical guidelines (Table 1) and that overall benefits of screening outweigh harm, which is yet to be proven in this setting 23.
- Blood test in CKD patients: there should be a commentary on the fact that troponin levels are commonly elevated in ESRD pts and that their evaluation over time should be monitored
We thank the reviewer for highlighting this pertinent point. We have added the following lines to highlight this important concept.
Lines 122 – 129: There are elevated baseline values of creatinine kinase (CK), creatinine kinase myocardial band (CK-MB) and cardiac troponin (cTn) in advanced CKD in the absence of acute coronary syndrome (ACS) 4,7. Regardless, elevated troponin T (TnT) and troponin I (TnI), both in the presence and absence of cardiac ischemia, are associated with increased all-cause and cardiovascular mortality in CKD and severe atherosclerotic CAD is more common among ESKD patients with elevated TnT 29.
In patients on dialysis, the sensitivity of high-sensitivity TnI for diagnosing MI remained high but specificity reduced 30. There is minimal variability in high-sensitivity TnT in stable dialysis patients so a routine test to establish a baseline TnT value could improve diagnosis of ACS31.
- The authors should point out more clearly what is new in their work as compared to former revies on this topic (see citation above also)
We acknowledge that literature has previously been produced on this topic but our review incorporates the results of the ISCHEMIA-CKD trial, the 2020 KDIGO guidelines, incorporates learning points from the JACC State-of-the-Art reviews on CKD and CAD, cardiac imaging for CAD risk stratification in CKD and the prognostic value of these investigations. We provide a comprehensive review on cardiovascular assessment and management including of CAD, arrhythmias, heart failure, pulmonary hypertension, valvulopathies and lifestyle factors.
Lines 37 – 47: Our review aims to serve as guide to clinicians on performing a comprehensive, wholistic cardiovascular screening assessment starting from the bedside assessment to utilization of novel, highly sophisticated screening methods such as cardiac PET. We provide a comprehensive review on cardiovascular assessment and management including for CAD, arrhythmias, heart failure, pulmonary hypertension, valvulopathies and lifestyle factors. We include a summary of the most recent 2020 Kidney Disease: Improving Global Outcomes (KDIGO) Clinical Practice Guideline4 and discuss the results of the ground breaking ISCHEMIA-CKD trial5. We discuss the value of each screening assessment especially in the context of CKD, controversies on pre-transplant cardiac screening, inconsistencies between guidelines and the advantages and limitations of different screening strategies with the aim to help the reader make informed decisions about pre-operative cardiac screening and cardiovascular disease management.
Round 2
Reviewer 2 Report
Dear Editor, thank you for the giving me the opportunity to review again the paper.
The main limitation of this article is that it touches on a very broad subject for which it is difficult to be exhaustive. Many aspects of the cardiac assessment may be the subjects of review by themselves. Thus, a review like this one may seem to address certain topics in a superficial manner.
However, the authors responded clearly to all comments and the paper is clearly improved.
Author Response
Dear Reviewer,
Thank you for your kind words and thorough review of our article. We completely agree that this is a very broad topic and it was not possible to cover each topic in the detail that we would have liked; however, we hope that our coverage of topics strikes a good compromise between breadth and comphrehensiveness.
Thank you once again for your kind comments.
Yours faithfully,
Madhivanan Elango
Reviewer 3 Report
Dear Editor
Unfortunately, despite the previous review, the authors continue their rhetoric. The sentence about the necessity of introducing a mechanism and criteria for patient selection cannot be written into contemporary transplantation standards [37-42].
The paper mainly devoted to the diagnostic possibilities of coronary artery disease. The chapters on smoking and obesity and the subsequent return to heart failure create a disorganized impression. There are no references to the most common dilemmas in KTx, such as hypertension and its perioperative treatment, anemia, the problem of high-vascular fistula and right ventricular overload syndrome, hemodynamic monitoring and the problem of vascular accesses, etc. The article frequently mentions renal dysfunction, but in practice this is far from the most common problems in KTx.
Reviewer 4 Report
The authors have largely improved their manuscript and answered adequately major issues raised by the reviewers. There are only some editiongn issues to be solved as:
"However, in the setting of transplantation, the argument that screening can guide decisions about transplant candidacy 6 96 and direct pretransplant optimisation."
Therefore, the authors are asked to thouroughly revise the final draft of their manuscript to sort out such minor editin issues, especially as the authors are native speakers.
Author Response
Dear Reviewer,
Thank you for your kind comments and thorough review of our paper. We have reviewed our paper thoroughly and made corrections (including to the highlighted sentence).
Thank you once again for your review of our paper.
Yours faithfully,
Madhivanan Elango